# Modality-specific tracking of attention and sensory statistics in the human electrophysiological spectral exponent

Leonhard Waschke[1,2]*, Thomas Donoghue[3], Lorenz Fiedler[4], Sydney Smith[5], Douglas D Garrett[1,2], Bradley Voytek[3,5,6,7]*†, Jonas Obleser[8,9]*†

[1]Max Planck UCL Centre for Computational Psychiatry and Ageing Research, Max Planck Institute for Human Development, Berlin, Germany; [2]Center for Lifespan Psychology, Max Planck Institute for Human Development, Berlin, Germany; [3]Department of Cognitive Science, University of California, San Diego, La Jolla, United States; [4]Eriksholm Research Centre, Oticon A/S, Snekkersten, Denmark; [5]Neurosciences Graduate Program, University of California, San Diego, La Jolla, United States; [6]Halıcıoglu Data Science Institute, University of California, San Diego, La Jolla, United States; [7]Kavli Institute for Brain and Mind, University of California, San Diego, La Jolla, United States; [8]Department of Psychology, University of Lübeck, Lübeck, Germany; [9]Center of Brain, Behavior, and Metabolism, University of Lübeck, Lübeck, Germany

*For correspondence:
waschke@mpib-berlin.mpg.de
(LW);
bvoytek@ucsd.edu (BV);
jonas.obleser@uni-luebeck.de
(JO)

†These authors share senior authorship to this work

**Abstract** A hallmark of electrophysiological brain activity is its 1/f-like spectrum – power decreases with increasing frequency. The steepness of this 'roll-off' is approximated by the spectral exponent, which in invasively recorded neural populations reflects the balance of excitatory to inhibitory neural activity (E:I balance). Here, we first establish that the spectral exponent of non-invasive electroencephalography (EEG) recordings is highly sensitive to general (i.e., anaesthesia-driven) changes in E:I balance. Building on the EEG spectral exponent as a viable marker of E:I, we then demonstrate its sensitivity to the focus of selective attention in an EEG experiment during which participants detected targets in simultaneous audio-visual noise. In addition to these endogenous changes in E:I balance, EEG spectral exponents over auditory and visual sensory cortices also tracked auditory and visual stimulus spectral exponents, respectively. Individuals' degree of this selective stimulus–brain coupling in spectral exponents predicted behavioural performance. Our results highlight the rich information contained in 1/f-like neural activity, providing a window into diverse neural processes previously thought to be inaccessible in non-invasive human recordings.

## Introduction

Non-invasive recordings of electrical brain activity represent the aggregated post-synaptic activity of large cortical neuronal ensembles (*Buzsáki et al., 2012*). Frequency spectra of electrophysiological recordings commonly display quasi-linearly decreasing power with increasing frequency (in log/log space; see *Figure 1*; see *Miller et al., 2009*; *Voytek et al., 2015*). In humans, this decrease is super positioned by several peaks, of which the most prominent is typically in the range of alpha oscillations (~8–12 Hz; see *Buzsáki et al., 2013*). The overall decrease in power as a function of frequency, f, reflects aperiodic as opposed to oscillatory activity; the steepness of such $1/f^\chi$ spectra can be captured by the spectral exponent $\chi$, wherein smaller values reflect flatter spectra.

Importantly, inter-individual differences in the steepness of human electroencephalography (EEG), power spectral densities (PSD), estimated by the spectral exponent $\chi$, are related to chronological age, behavioural performance, and display stable inter-individual differences (*Dave et al., 2018*; *Donoghue et al., 2020*; *Sheehan et al., 2018*; *Voytek et al., 2015*; *Waschke et al., 2017*). Intra-individual variations in EEG spectral exponents have been reported as a function of overall arousal level and activation (*Colombo et al., 2019*; *Lendner et al., 2020*; *Podvalny et al., 2015*). Based on computational models and invasive recordings of neural activity, it has been demonstrated that electrophysiological spectral exponents capture the balance of excitatory and inhibitory neural activity (E:I; *Gao et al., 2017*), with recent causal, optogenetic work showing that lower exponents are mechanistically linked to increased E:I balance (*Chini et al., 2021*). Although unknown at present, it is plausible that differences in non-invasive electrophysiological spectral exponents might also reflect variations in E:I balance. Additionally, it remains unclear if specific, behaviourally relevant intra-individual E:I variations that accompany the allocation of attentional resources in non-human animals (*Ferguson and Cardin, 2020*; *Kanashiro et al., 2017*; *Ni et al., 2018*) can be inferred in a similar manner in humans. An attentional shift towards a given sensory modality is suggested to entail desynchronized activity (i.e., reduced low-frequency oscillations and increased high-frequency power) in cortical areas that process information from the currently attended domain (*Cohen and Maunsell, 2011*; *Harris and Thiele, 2011*). These shifts in desynchronization likely trace back to an attention-related change in E:I balance towards excitation (*Harris and Thiele, 2011*; *Zagha and McCormick, 2014*), which is thought to manifest in a reduction of spectral exponents (*Gao et al., 2017*; *Waschke et al., 2019*). If EEG spectral exponents indeed represent a sensitive approximation of E:I balance, selective attention should also result in a topographically specific decrease of exponents. Such attention-related decreases in EEG spectral exponents should take place over and above potential changes in alpha oscillations which have long been assumed to capture dynamics of sensory cortical inhibition in the service of selective attention (*Jensen and Mazaheri, 2010*; *Klimesch et al., 2007*).

A conjecture of the present study thus is that 1/f-like EEG activity captures changes in the E:I balance of underlying neural populations. Such a non-invasive approximation of variations in human E:I would be of great value, enabling investigations of processes previously inaccessible using non-invasive imaging techniques. This includes the role of E:I in sensory processing and perception (*Wehr*

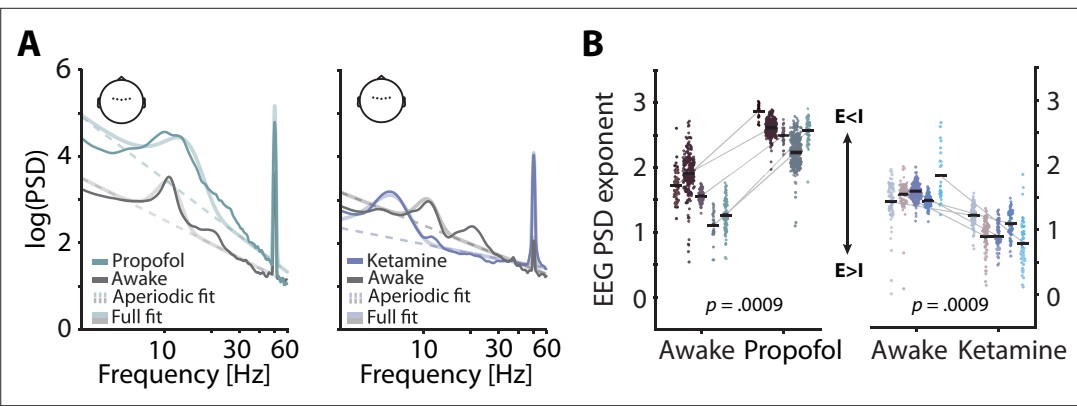

**Figure 1.** Electroencephalography (EEG) spectral exponents track anaesthesia-induced E:I changes. (**A**) Normalized EEG spectra averaged across five subjects and five central electrodes (inset) displaying a contrast between rest and propofol (left) and ketamine anaesthesia (right). Spectral parameterization yielded aperiodic fits that estimated the spectral exponent (dashed lines) and full fits that included oscillatory spectral peaks (transparent lines). (**B**) Pairwise scatter plots depicting subject-wise averaged EEG spectral exponents during awake rest, propofol (left) and ketamine (right). Coloured dots represent spectral exponents of 5 s snippets, black horizontal bars single subject means. p-Values based on 1000 random permutations of data.

The online version of this article includes the following figure supplement(s) for figure 1:

**Figure supplement 1.** Raw EEG power spectra during awake rest, ketamine, and propofol anaesthesia.

**Figure supplement 2.** Topographically-resolved t-statistics comparing EEG spectral exponents between awake rest and different anaesthetics.

*and Zador, 2003*; *Wörgötter et al., 1998*), selective attention (*Harris and Thiele, 2011*) and ageing (*Luebke et al., 2004*), and not least in disease (*Cummings et al., 2009*; *Dani et al., 2005*; *Lisman, 2012*).

We here will demonstrate the sensitivity of EEG spectral exponents to variations in E:I in two ways. First, predicated on the differential effects of propofol and ketamine (*Concas et al., 1991*; *Deane et al., 2020*; *Franks, 2008*) on E:I (im-)balance, and building on invasive as well as modelling work, we compare EEG spectra of a previously published dataset (*Sarasso et al., 2015*) between quiet wakefulness, propofol, and ketamine anaesthesia and demonstrate that EEG spectral exponents are valid markers of broad intra-individual variations in E:I balance. Considering the EEG spectral exponent as a viable estimate of E:I balance, we then sought to test the extent to which it could be modulated also through a cognitive manipulation.

Second, we investigated the critical role of E:I balance in attention. We analysed data from human participants who performed a challenging multisensory detection task during which they had to attend to one of two concurrently stimulated sensory modalities (auditory vs. visual) to detect faint target stimuli. Our results demonstrate that despite constant multisensory input, selective attention entails a modality-specific reduction of spectral exponents (spectral flattening) that is in line with an increased E:I ratio.

When aiming to explain the 1/f spectral exponent of the EEG and its relation to E:I balance, it is important to also consider potential other factors that might alter spectral exponents. This is relevant especially since 1/f-like processes are not limited to the brain, but are ubiquitous in nature (*Brown et al., 2002*; *Coensel et al., 2003*; *Keshner, 1982*; *Mandelbrot and Wheeler, 1983*). Given reports of behaviourally relevant synchronization of oscillatory brain activity with oscillatory sensory inputs (*Breska and Deouell, 2017*; *Henry et al., 2014*; *Lakatos et al., 2008*; *Spaak et al., 2014*), this raises the question of whether a similar link might exist between the 1/f profiles of neural responses and sensory inputs. Indeed, even speech signals show a pronounced 1/f shape within their amplitude modulation (AM) spectrum (see e.g., *Attias and Schreiner, 1997*). Hence, 1/f-like sensory inputs might hold ecological relevance. However, despite first non-human in vitro and in vivo evidence for such a neural tracking of 1/f sensory signals (*Nozaki et al., 1999*; *Qu et al., 2019*; *Yu et al., 2005*), it is unclear at present if the 1/f structure of sensory inputs is tracked by 1/f-like non-invasively recorded neural activity in humans.

How would 1/f characteristics of sensory input map onto 1/f of neural activity, if not by changing endogenous E:I balance? Most parsimoniously, a spectral-exponent concordance between sensory statistics and EEG activity might trace back to the superposition of postsynaptic potentials, the main source of EEG signals (*Buzsáki et al., 2012*). Postsynaptic potentials in sensory cortical areas could temporally align and scale with the magnitude of sensory stimuli and hence result in a mimicking of the stimulus spectrum, as has been suggested for oscillatory signals and steady state evoked potentials (e.g., *Norcia et al., 2015*) and broadband signals or speech (*Lalor et al., 2009*). It thus appears plausible that the spectral exponent of the human brain will capture and prove not only endogenous changes in E:I balance but also reflect statistical features of the sensory input. If such non-oscillatory neural tracking of 1/f sensory features (i.e., an alignment of neural and sensory spectral exponents) is non-invasively detectable in humans, this would greatly extend the toolset of perceptual neuroscience to including a wide range of naturalistic (1/f-like) stimuli.

Thus, in addition to neurochemical and attentional effects on EEG spectral exponents that likely capture changes in E:I balance, we investigated the link between environmental 1/f input and 1/f-like brain activity. We tested the single trial relationship between 1/f sensory features and 1/f EEG spectral exponents. Revealing modality-specific and topographically distinct links of stimulus exponents and EEG spectral exponents that explain inter-individual differences in task performance, we present evidence for the behaviourally-relevant neural tracking of 1/f-like sensory features.

## Results
### Establishing the validity of the EEG spectral exponent as a non-invasive marker of E:I

Previous invasive animal work has demonstrated the sensitivity of spectral exponents to anaesthesia-related changes in the balance of excitatory and inhibitory neural activity (E:I balance; *Gao et al.,*

*2017*). To expand these results and test, the sensitivity of non-invasively recorded human EEG to physiological changes in E:I balance, we contrasted spectral exponents of human EEG recordings between quiet wakefulness and anaesthesia for two different general anaesthetics: propofol and ketamine. Both anaesthetics exert widespread effects on the overall level of neural activity (*Taub et al., 2013*) as well as on oscillatory activity in the range of alpha and beta (8–12 Hz; ~ 15–30 Hz). Importantly, however, propofol is known to commonly result in a net increase of inhibition (*Concas et al., 1991*; *Franks, 2008*) whereas ketamine results in a relative increase of excitation (*Deane et al., 2020*; *Miller et al., 2016*). In accordance with invasive work and single cell modelling (*Chini et al., 2021*; *Gao et al., 2017*), propofol anaesthesia should thus lead to an increase in the spectral exponent (steepening of the spectrum) and ketamine anaesthesia to a decrease (flattening). Based on previous results, the effect of anaesthesia on EEG spectral exponents is expected to be highly consistent and display little topographical variation (*Lendner et al., 2020*). For simplicity, we focused on a set of five central electrodes that receive contributions from many cortical and subcortical sources (see *Figure 1*) but report topographically resolved effects in the supplements (see *Figure 1—figure supplement 2*). Here, propofol anaesthesia led to an overall increase in EEG power which was especially pronounced in the alpha-beta range. Ketamine anaesthesia decreased the frequency of alpha oscillations and supressed power in the beta range. Importantly, however, EEG spectral exponents that were estimated while accounting for changes in oscillatory activity increased under propofol and decreased under ketamine anaesthesia in all participants (both $p_{permuted}$ < 0.0009, *Figure 1*). These results replicate previous invasive findings and support the validity of EEG spectral exponents as markers of overall E:I balance in humans.

## Can the EEG spectral exponent track attentional focus and sensory statistics?

The findings outlined above demonstrate the ability of the EEG spectral exponent to non-invasively track intra-individual changes in E:I balance. Under the presumption that the spectral exponent can thus be used as a viable marker of E:I balance in general, we then sought to test the extent to

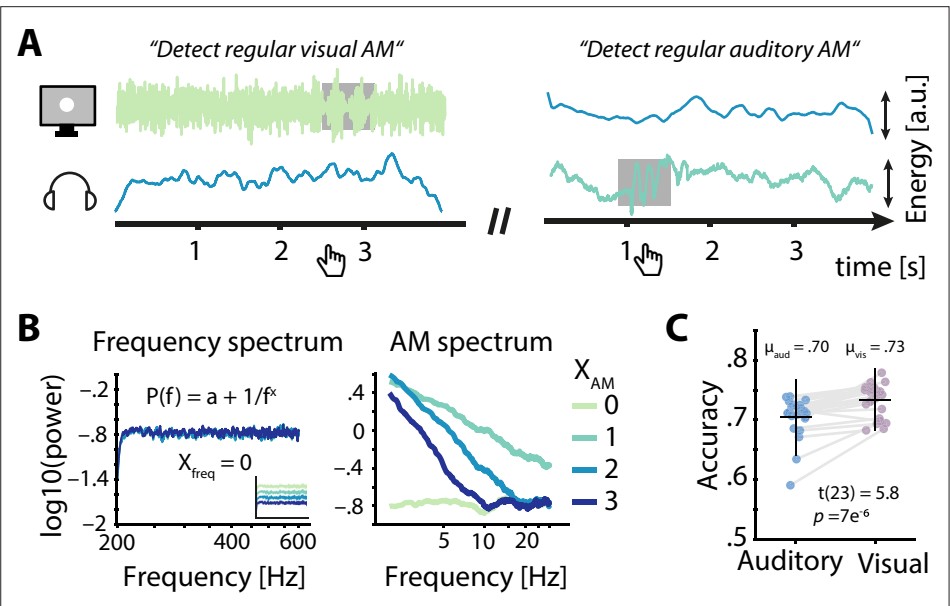

**Figure 2.** Task design and behavioural performance. (**A**) Participants were simultaneously presented with auditory and visual amplitude modulated (AM) noise and had to detect periods of sinusoidal AM (grey box) in the luminance variations of a visually presented disk (left) or in auditory presented white noise (right) by pressing a button. Example visual stimuli correspond to spectral exponents of 0 (left) and 2 (right), auditory stimuli to spectral exponents of 2 (left) and 1 (right). (**B**) Frequency spectra for four sets of AM spectra (left) demonstrate the identical flat spectra (white noise), further visualized by artificially offset spectra in the inset. AM spectra displayed spectral exponents between 0 and 3 (right). (**C**) Auditory accuracy (70 %) was significantly lower than visual accuracy (73 %). Dots represent single subject data, horizontal lines the mean, and vertical lines ± one standard error.

which the spectral exponent could be modulated through cognitive manipulation, rather than through pharmacological intervention. Accordingly, using an audio-visual decision-making task, we examined selective attention-related variations in the EEG spectral exponent. In addition, expanding on the idea of 1/f-like sensory signals being tracked by cortical activity, we used these data to test a link between the spectral exponents of audio-visual stimuli and recorded EEG signals.

## Behavioural performance in a multimodal detection task

Participants (N = 24) performed a challenging multisensory task during which participants had to detect brief time periods during which the AM of the presented white noise switched from aperiodic to sinusoidal (*Figure 2A*). In detail, participants attended either auditory or visual noise stimuli, which were always presented simultaneously and displayed AM spectra with spectral exponents between 0 and 3 (*Figure 2B*). While training and adaptive adjustments of difficulty (see methods for details) ensured that the task was challenging but doable in both modalities (average accuracy $\geq$ 70 %), participants performed better during visual compared to auditory trials ($t_{23}$ = 5.8, p = 7 × $10^{-6}$, Cohen's d = 1.18; *Figure 2C*).

## Modality-specific attention selectively reduces the EEG spectral exponent

To further explore the sensitivity of EEG spectral exponents to specific experimental manipulations, we investigated changes in selective attention, which have been proposed to coincide with a relative increase of excitatory neural activity in sensory cortices of the attended domain. Average EEG spectral exponents for central and occipital regions of interest (see insets in *Figure 3*) were controlled for neural alpha power and a set of additional nuisance variables (see Methods for details and control analyses for a separate analysis of alpha power) and compared between auditory and visual attention using a 2 × 2 repeated measures analysis of variance. In addition to a main effect of attentional focus ($F_{1,92}$ = 12.98, p = 0.0005, partial eta squared = 0.124), this analysis revealed an interaction between attentional focus and ROI ($F_{1,92}$ = 12.91, p = 0.0005, partial eta squared = 0.123). Notably, EEG spectral exponents at occipital electrodes strongly decreased (spectra flattened) under visual compared to auditory attention ($t_{23}$ = 7.4, p = 1 × $10^{-7}$, Cohen's d = 1.52; see *Figure 3B*), while this was the case to a much lesser extent at central electrodes ($t_{23}$ = 2.6, p = 0.01, Cohen's d = 0.54).

The topographical specificity of this attention-induced spectral flattening was qualitatively confirmed by a cluster-based permutation test on the relative (z-scored) single-subject EEG spectral exponent differences between auditory and visual attention, revealing a central, negative cluster (p = 0.04) and an occipital positive cluster (p = 0.01). Thus, the selective allocation of attentional resources to one modality results in a flattening of the EEG power spectrum over electrodes typically associated with this modality, especially for occipital electrodes during visual attention.

## 1/f-like stimulus properties are tracked by modality-specific changes in EEG spectral exponent

Next, our goal was to delineate how changes in the EEG spectral exponent might be driven by sensory input statistics, instead of endogenous changes to E:I balance. In addition to attention-related variations in brain dynamics, we probed the link between spectral exponents of sensory stimuli and EEG activity. This approach departs from the focus on EEG spectral exponents as a non-invasive approximation of E:I and aims to test how the alignment of sensory input with sensory cortical activity could shape EEG spectral exponents to mimic the spectra of sensory signals.

Electrode-wise linear mixed effect models of EEG spectral exponents were used to test the relationship between the AM spectral exponents of presented stimuli and recorded EEG activity (see model details in *Supplementary file 1*). Of note, the impact of early sensory-evoked responses on estimates of 1/f stimulus tracking (i.e., the linear relationship between EEG spectral exponents and AM stimulus spectral exponents) was excluded by limiting analyses to the time period between 600 m s after stimulus onset and target onset. Single trial parameterizations of power spectra provided excellent fits for the vast majority of trials (mean $R^2$ > 0.84 at all electrodes, see *Figure 1—figure supplement 2b*). Note that reliable spectral parameterization is key to disentangle oscillatory from aperiodic activity, enabling us to analyse EEG spectral exponents which otherwise would be confused with a mix of low frequency and high frequency power. A main effect of auditory stimulus exponent,

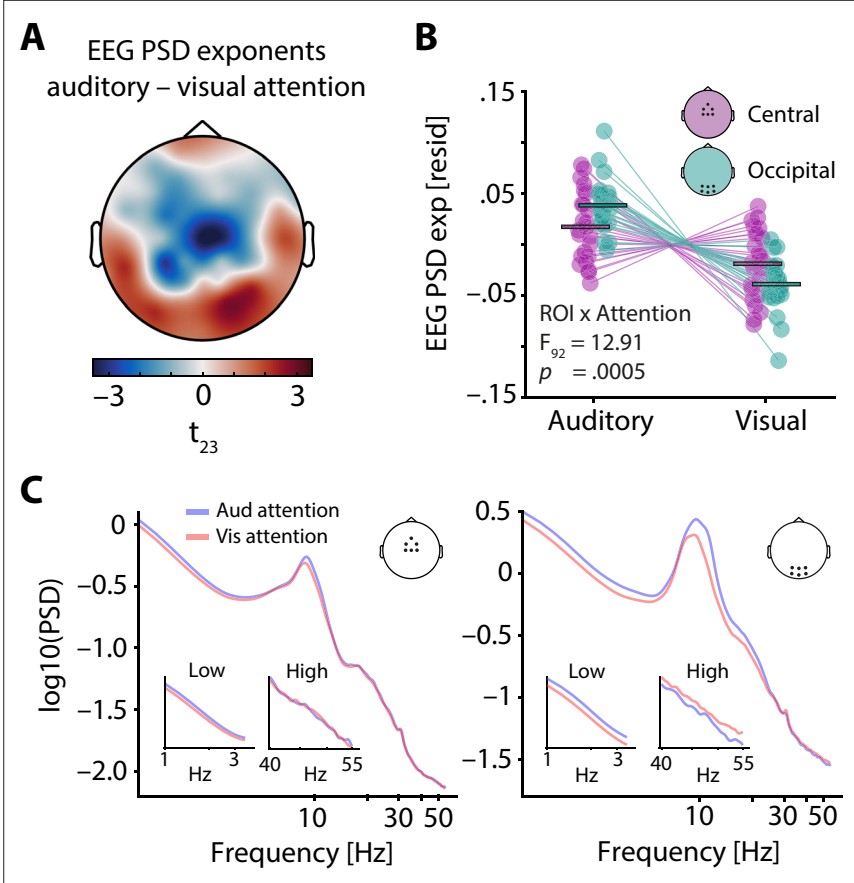

**Figure 3.** Electroencephalography (EEG) spectral exponents track the focus of selective attention. (**A**) t-values of the average difference between EEG spectral exponents during auditory and visual attention. Note that exponents were controlled for neural alpha power and other confounding variables before subject-wise differences were calculated. A central negative and an occipital positive cluster are clearly visible. (**B**) Average central (lilac) and occipital EEG spectral exponents (teal) for auditory and visual attention (residuals shown). Horizontal bars denote the grand average. While visual attention as compared to auditory attention was associated with a decrease of spectral exponents at central and occipital sites, this decrease was more pronounced at occipital electrodes. This interaction of ROI × Attention is captured by the cross-over of lilac and teal lines. (**C**) Grand average spectra for auditory (blue) and visual attention (red), shown for a central (left) and an occipital ROI (right). Insets display enlarged versions of spectra for low and high frequencies, separately.

The online version of this article includes the following figure supplement(s) for figure 3:

**Figure supplement 1.** Evoked responses (ERPs) as a function of attentional focus.

capturing the trial-wise positive linear relationship between auditory AM spectral exponents and EEG spectral exponents (i.e., neural tracking) was found at a set of four central electrodes (all z > 3.5, all $p_{corrected}$ < 0.02; see *Figure 4A*). Similarly, a positive main effect of visual stimulus exponent was present at a set of three occipital electrodes (all z > 3.7, all $p_{corrected}$ < 0.01; see *Figure 4A*). Hence, EEG spectral exponents displayed a topographically resolved tracking of stimulus exponents of both modalities. Additionally, standardized single subject estimates of stimulus tracking at both electrode clusters identified by the mixed model approach were extracted after controlling for covariates also used in the final mixed model and revealed qualitatively similar results for both auditory ($t_{23}$ = 5.1, p = 0.00004, Cohen's d = 1.03) and visual stimulus tracking ($t_{23}$ = 3.9, p = 0.0008, Cohen's d = 0.79; see *Figure 4A*).

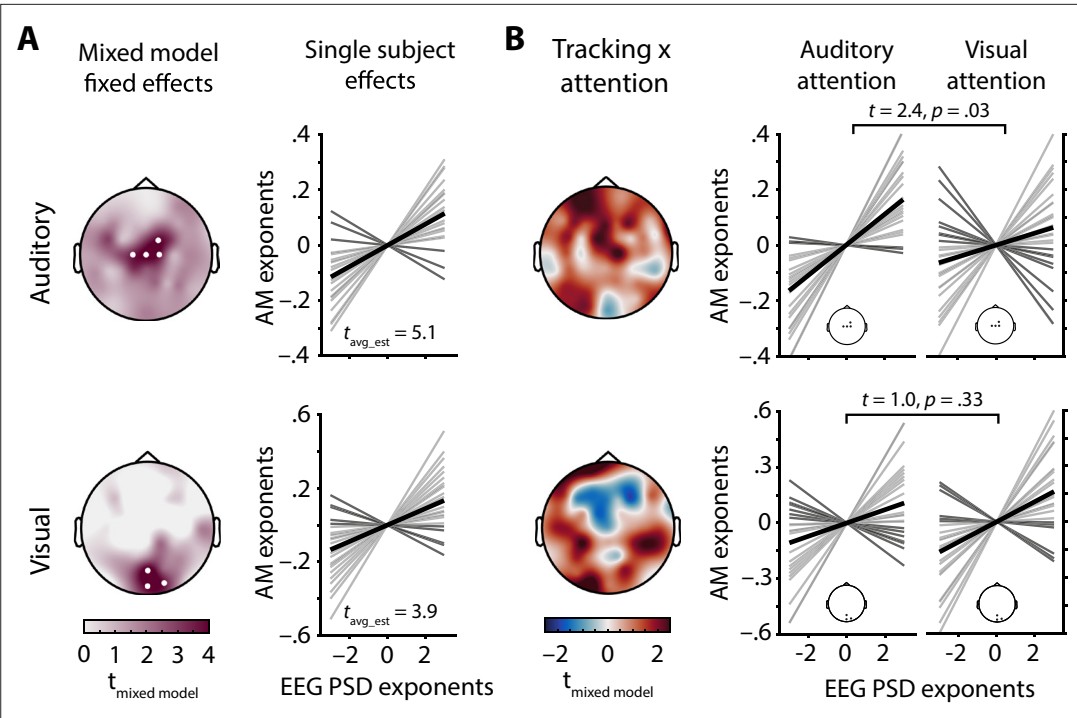

**Figure 4.** Electroencephalography (EEG) spectral exponents track stimulus spectral exponents. (**A**) Topographies depict t-values for the main effect of stimulus spectral exponent, taken from a mixed model of EEG spectral exponents. White dots represent electrodes with significant effects after Bonferroni correction. Auditory stimulus tracking (upper row) clusters at central electrodes, visual stimulus tracking (lower row) at occipital electrodes. Line plots show single subject tracking estimates (standardized betas), dark grey lines represent negative betas, black lines the grand average. (**B**) Topographies show t-values for the interaction of attentional focus and stimulus spectral exponent, taken from the same mixed model as in A. Positive clusters appear over central (auditory) and occipital (visual) areas and represent improved tracking during selective attention to the modality in question. Line plots visualize single subject effects of selective attention on stimulus tracking for the clusters found in A (insets). Note that t-values for the auditory stimulus × attention interaction were inverted to remove the sign change caused by zero-centred effect coding of attentional focus. In this way, positive t-values represent evidence for an increase in stimulus tracking if attention is directed towards this modality.

The online version of this article includes the following figure supplement(s) for figure 4:

**Figure supplement 1.** Evoked responses (ERPs) as a function of stimulus spectral exponents.

**Figure supplement 2.** Trial-wise spectral parameterization fit statistics and examples.

**Figure supplement 3.** Grand average auditory (top) and visual (bottom) temporal response functions for auditory (blue) and visual attention (red).

**Figure supplement 4.** Overview of backward model performance.

**Figure supplement 5.** Predicted Electroencephalography (EEG) spectra.

**Figure supplement 6.** Model comparison topographies.

**Figure supplement 7.** Mixed model results, controlling for single trial performance.

## Stimulus tracking in EEG spectral exponent interacts with attentional focus

To further investigate how the just-observed neural tracking of stimulus spectral exponents might interact with the putatively E:I-balance-mediated changes in attentional focus, we tested for stimulus exponent× attention interactions on the observed EEG spectral exponent, within a mixed-model framework.

The interaction of auditory stimulus exponents and attentional focus surfaced as a positive central cluster, and the interaction of visual stimulus exponents and attentional focus yielded a positive occipital cluster (see *Figure 4B*).

As can be discerned from the topographies in *Figure 4B*, selective attention likely improved the tracking of stimulus spectral exponents over sensory-specific areas, separately for each sensory domain. To extract single subject estimates of the stimulus × attention interaction, we controlled for several covariates, focusing on EEG spectral exponents averaged within the clusters that displayed significant tracking (see above). Paired t-tests comparing standardized regression coefficients that represent the strength of neural stimulus tracking revealed a significant increase of auditory stimulus tracking under auditory attention ($t_{23}$ = 2.4, p = 0.03, Cohen's d = 0.49). Visual stimulus tracking did not significantly improve under visual attention, though the direction of the effect was the same ($t_{23}$ = 1.0, p =0 .33, Cohen's d = 0.20). Thus, modality-specific attention was selectively associated with an improvement in the neural tracking of auditory AM stimulus spectra.

## Evoked responses and spectral power estimates do not index the spectral exponent of audiovisual stimuli

To test whether conventional estimates of sensory evoked EEG activity tracked the spectral exponent of stimuli, we analysed evoked potentials (ERPs) at the single-trial level. This analysis did not reveal significant ERP clusters for either auditory (p > 0 .3) or visual (p > 0.2) spectral AM tracking (see *Figure 4—figure supplement 1* for ERPs). The absence of such effects is visualized in *Figure 4—figure supplement 4* which displays ERPs time-courses for four bins of increasing auditory as well as visual AM spectral exponents. In contrast to 1 /f EEG exponents, conventional metrics of sensory evoked EEG activity were thus insensitive to the AM spectral exponents of presented stimuli.

Furthermore, to estimate the specificity of the link between EEG spectral exponents and stimulus exponents, we also tested the tracking of stimulus exponents within low-frequency (1–5 Hz) and alpha power (8–12 Hz). To this end, we inverted linear mixed models to either include auditory or visual stimulus exponent as the dependent variable and low-frequency power, alpha power, or EEG spectral exponents as predictors of interest (covariates kept constant across models). Model comparisons revealed better model fits based on EEG spectral exponents throughout (see *Figure 4—figure supplement 6*). Only when modelling visual stimulus exponents, alpha power explained significantly more variance than EEG spectral exponents at one parietal electrode that was not part of the significant tracking cluster reported above (*Figure 4A*).

## The extent of modality-specific spectral-exponent tracking predicts behavioural performance

Next, to investigate the link between individual levels of neural stimulus tracking and behavioural performance, we computed between-subject correlations of standardized stimulus tracking betas (auditory and visual) and common metrics of behavioural performance (accuracy and response speed, separately for both modalities) using a multivariate partial least squares analysis (PLS; see Methods for details; *McIntosh et al., 1996*). This model revealed one significant latent variable (p = 0.03) that captured the low-dimensional latent space of the correlation between neural stimulus tracking and behavioural performance, which displayed a clear fronto-central topography (*Figure 5A*).

At the overall latent level, both auditory (rho = 0.49, p = 0.016) and visual stimulus tracking (rho = 0.43, p = 0.035) were significantly correlated with performance across subjects (*Figure 5B*). In detail, auditory stimulus tracking was positively correlated with auditory response speed (rho = 0.30, CI = [0.03, 0.64]) but negatively correlated with visual accuracy (rho = –0.28, CI = [–0.62, –0.06]) and response speed (rho = –0.36, CI = [–0.73, –0.17]). Analogously, visual stimulus tracking was positively linked with visual accuracy (rho = 0.31, CI = [0.15, 0.68]) and response speed (rho = 0.35, CI = [0.11, 0.72]; see *Figure 5A*), but showed no significant relationship with auditory performance.

Taken together, participants who displayed stronger neural tracking of AM stimulus spectra also performed better. However, the behavioural benefit of stimulus tracking was modality-specific: performance in one sensory domain (e.g., auditory) only benefited from tracking within that domain, but not in the other (e.g., visual). Instead, visual detection performance was slower and less accurate in individuals who displayed strong neural tracking of 1/f-like auditory stimulus features.

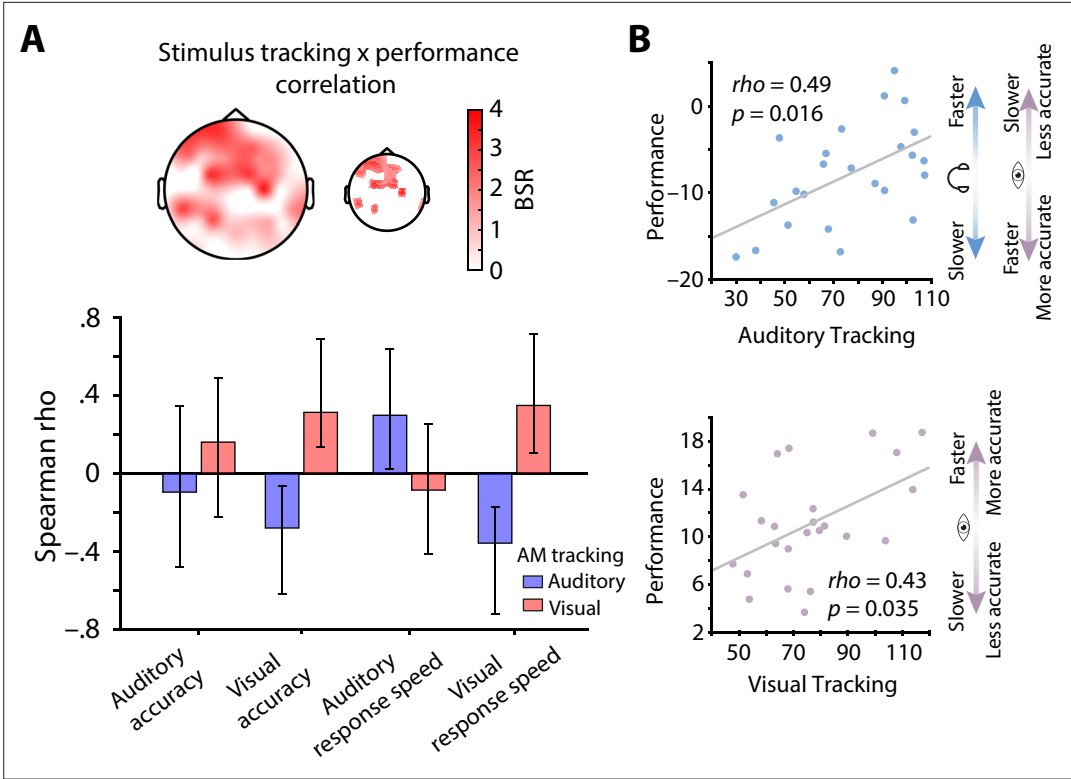

**Figure 5.** Stimulus tracking explains inter-individual differences in performance. (**A**) Results of a multivariate neuro-behavioural correlation between stimulus tracking and performance using partial least squares analysis (PLS). The topography depicts bootstrap ratios (BSR) of the first latent variable and can be interpreted as z-values. Within the smaller topography, BSRs are thresholded at a BSR of 2 (p < 0.05). Bar graphs represent the correlation (Spearman rho) between auditory (blue) and visual stimulus tracking (red) with performance (accuracy as % correct and response speed; RS in s–1) in both modalities. Vertical lines denote 95 % bootstrapped confidence intervals. (**B**) Scatter plots of latent correlations between latent auditory (upper panel) and visual tracking (lower panel) with latent performance, respectively. Auditory stimulus tracking was positively linked with auditory performance but negatively linked with visual performance. Visual stimulus tracking was positively linked with visual performance. Headphones and eye symbolize auditory and visual performance, respectively.

## Control analysis: no stimulus-exponent effects on evoked potentials or oscillatory power

To compare established metrics of early sensory processing, we contrasted evoked EEG activity of different attention conditions. Event-related potentials (ERPs) after noise onset were compared between auditory and visual attention for electrodes Cz and Oz, separately. While there was no significant attention-related ERP difference found at electrode Cz, early ERP components at electrode Oz were increased during visual attention (~.08 –.15 s post-stimulus, $p_{corrected}$ < 0.05; see ***Figure 3—figure supplement 1***).

Also, we subjected single trial alpha power (after partialing for EEG spectral exponents) from the ROIs shown in ***Figure 3*** to the same ANOVA (Attention (2) × ROI (2)) as EEG spectral exponents. This analysis revealed a main effect of attention ($F$ = 20.3, p = 0.00002, partial eta squared = 0.180) as well as an interaction of attention and ROI ($F$ = 20.4, p = 0.00002, partial eta squared = 0.181). Hence, alpha power increased from visual to auditory attention in a region-specific manner, with stronger increases over occipital cortical areas (see spectra in ***Figure 3C***).

Importantly, this attention-related change in alpha power took place over and above the observed attentional modulation of EEG spectral exponents for at least three reasons: First, spectral parameterization separates oscillatory and non-oscillatory components, dissecting alpha power from the spectral exponent (***Donoghue et al., 2020***). Second, and to account for shared variance between alpha power and EEG spectral exponents that might occur due to fitting issues, each measure had been partialed out from each other to account for collinear contributions of the two. Third, attention-related

modulations of alpha power and EEG spectral exponents displayed no significant correlation across participants (all electrode-wise $p_{FDR} > 0.15$).

The presence of modality-specific, attention-dependent changes despite these controls strongly suggests that neural alpha power, as well as EEG spectral exponents, track distinct changes in neural activity that accompany attentional shifts, but they are not affected by the spectral content of the stimuli themselves.

## Exploratory analysis: estimating temporal response functions to link stimulus spectra and resulting EEG spectra

To directly test to what degree spectral exponent changes in the stimulus material become reflected in the spectral exponent of the EEG response via a time-resolved, evoked-response–like 'neural tracking' (*Obleser and Kayser, 2019*), we also employed an alternative analysis framework to estimate the link between sensory input and electrophysiological activity in the time domain. Specifically, we used temporal response functions (TRF; *Crosse et al., 2016*; *Lalor and Foxe, 2010*; *Wöstmann et al., 2016*) to capture the temporally resolved multivariate linear relationship between EEG and continuous sensory input.

After estimating a modality-specific impulse response function (see *Figure 4—figure supplement 3*), this function can be used to either reconstruct stimulus input (backward model) or predict neural recordings (forward model) from the combination of estimated TRFs and EEG data or stimulus time series. Importantly, auditory response functions appeared unreliable and displayed an uncommon shape with only one peak at a lag of 200 ms whereas visual response functions appeared stable. In line with his observation, backward models only yielded significant stimulus reconstruction for the visual domain (see *Figure 4—figure supplement 4*). Given the absence of reliable auditory TRFs and backward model predictions, we modelled EEG time-series based on a canonical TRF (grand average visual TRF) and stimulus data. This simplistic model indeed produced EEG data whose spectral exponents were positively linked with the spectra of presented stimuli (*Figure 4—figure supplement 5*) both for the auditory ($\beta = 0.37$, SE = 0.08, t = 4.3, p < 0.0001) and the visual domain ($\beta = 0.59$, SE = 0.07, t = 8.9, p = 0).

Hence, while TRF-based models were unable to empirically demonstrate a significant link between 1/f-like stimulus input and 1/f-like EEG activity given the data analysed in the current manuscript, these results illustrate the general feasibility and potency of such an approach. However, future work is needed to empirically demonstrate that a positive link between the spectral exponents of sensory input and EEG activity traces back to the temporal alignment of both. Overall, and in given our data, a spectrally based approach that directly links spectra of presented stimuli with those of recorded EEG appears as the more powerful approach to detect an alignment of input and neural activity.

## Discussion

We have presented evidence for the sensitivity of the EEG spectral exponent to a variety of influences — neurochemical, cognitive, and sensory in nature. Jointly, these results underscore the value of 1/f-based neural measures such as the spectral exponent for studying the neural bases of perception and behaviour. Specifically, we here have shown that the EEG spectral exponent (1) reflects systemic, anaesthesia-induced changes in brain state closely linked to E:I balance, (2) captures focal attention-related changes in E:I brain state, and (3) simultaneously is able to track sensory stimuli by way of stimulus-specific spectral exponents. Furthermore, when presenting stimuli in the auditory and visual domain in parallel, stimulus tracking retained its modality-specificity in the spatial distribution of EEG spectral exponents. Not least, EEG spectral exponents explained inter-individual variance in behavioural performance, underlining the functional relevance of 1 /f processes in human brain activity.

### EEG spectral exponents as a non-invasive approximation of E:I balance

As hypothesized, our initial analysis of a general-anaesthesia dataset confirmed that propofol leads to steeper EEG spectra, while ketamine causes flattening (see *Figure 1*). Importantly, the spectral parameterization approach used here enabled us to separate changes in spectral exponents from other prominent anaesthesia-driven alterations such as boosted oscillatory power in the alpha-beta range (propofol) or decreased alpha frequency (ketamine). While these changes in oscillatory activity might

capture anaesthesia-related effects on connectivity and link to the loss of consciousness (*Purdon et al., 2013*), the present study focused on changes in EEG spectral exponents with the goal of evaluating its sensitivity to changes in E:I balance. The observed increase of spectral exponents during propofol anaesthesia is generally in line with previous results that relied on different measures and methodologies (*Colombo et al., 2019*; *Waschke et al., 2019*). Increased spectral exponents under propofol likely trace back to strengthened inhibition via increased activity at gamma-Aminobutyric acid receptors (GABAergic activity) and hence a reduced E:I balance as compared to quiet wakefulness (*Concas et al., 1991*; *Franks, 2008*). Furthermore, ketamine-induced decreases of spectral exponents likely depict the outcome of overall decreased inhibition that is caused by the blocking of excitatory N-methyl-D-aspartate (NMDA) receptors and an associated decrease in GABA release, resulting in an increased E:I balance (*Behrens et al., 2007*; *Deane et al., 2020*; *Franks, 2008*).

Our results build on the findings of previous studies (*Gao et al., 2017*; *Lendner et al., 2020*; *Medel et al., 2020*); replicate and validate these earlier results in non-invasive recordings; and offer crucial advancements by directly comparing the effect of two distinct anaesthetics on EEG spectral exponents. Hence, EEG spectral exponents pose a non-invasive approximation of intraindividual state changes in E:I balance.

While the EEG spectral exponent as a remote, summary measure of brain electric activity can obviously not quantify local E:I in a given neural population, the non-invasive approximation demonstrated here enables inferences on global neural processes previously only accessible in animals and using invasive methods. Future studies should use a larger sample to directly compare dose-response relationships between GABA-A agonists or antagonists (e.g., Flumanezil) and the EEG spectral exponent as well as common oscillatory changes.

Next, we capitalized on the sensitivity of the EEG spectral exponent to broad changes in E:I by investigating its ability to capture attention-related fluctuations in cortical E:I balance in a dedicated EEG experiment.

## EEG spectral exponents track modality-specific, attention-induced changes in E:I

While the EEG spectral exponent here proved sensitive to systemic, exogenously driven, drug-induced alterations in E:I, we also sought to test whether it can also be leveraged to detect non-drug-induced but endogenous, attention-related variations in E:I balance. Invasive animal work suggests that the allocation of attentional resources should result in a topographically specific shift towards desynchronized activity and an increased E:I ratio (*Harris and Thiele, 2011*; *Kanashiro et al., 2017*).

Here, we observed a topographically-specific pattern of reduced EEG spectral exponents through modality-specific attention. As indexed by a significant interaction between region of interest and attentional focus, visual attention led to a flattening of EEG spectra that was especially pronounced over occipital EEG channels (*Figure 3*). The absence of a comparable effect for auditory attention at central electrodes potentially traces back to the varying sensitivity of EEG recordings to different cortical sources. Central electrodes capture auditory cortical activity but are positioned far away from their dominant source ( *Huotilainen et al., 1998*; *Stropahl et al., 2018*). Occipital electrodes, however, are sensitive to visual cortex activity and are directly positioned above it (*Hagler et al., 2009*). Further exaggerated by the scaling of volume conduction with distance, this likely results in a reduced signal to noise ratio for auditory compared to visual cortex activity (*Piastra et al., 2020*).

Despite these differences in the sensitivity of EEG signals, our results provide clear evidence for a modality-specific flattening of EEG spectra through the selective allocation of attentional resources. This attention allocation likely surfaces as subtle changes in E:I balance (*Börgers et al., 2005*; *Harris and Thiele, 2011*). Importantly, these results cannot be explained by observed attention-dependent differences in neural alpha power (8–12 Hz, *Figure 3*), which have been suggested to capture cortical inhibition or idling states (*Cooper et al., 2003*; *Pfurtscheller et al., 1996*). The employed spectral parameterization approach enabled to us to separate 1 /f like signals from oscillatory activity and hence offered distinct estimates of spectral exponent and alpha power that would otherwise have been conflated (*Donoghue et al., 2020*). By accounting for the shared variance of EEG spectral exponents and alpha power while modelling the impact of modality-specific attention, we further controlled for a potential conflation of both and present evidence for their distinct involvement in the allocation of attentional resources.

How could attentional goals come to shape spectral exponents and alpha oscillations? Both attention-related changes in EEG activity might trace back to distinct functions of thalamo-cortical circuits. On the one hand, bursts of thalamic activity that project towards sensory cortical areas might sculpt cortical excitability in an attention-dependent manner by inhibiting irrelevant distracting information (*Klimesch et al., 2007*; *Saalmann and Kastner, 2011*). On the other hand, tonic thalamic activity likely drives cortical desynchronization via glutamatergic projections and, with attentional focus, results in boosted representations of stimulus information within brain signals (*Cohen and Maunsell, 2011*; *Harris and Thiele, 2011*; *Sherman, 2001*).

Our findings of separate attentional modulations of both, EEG spectral exponents and alpha power, point towards the involvement of both thalamic modes in the realization of attentional states. Recently, momentary trade-offs between both modes of thalamic activity have been suggested to give way to attention-related modulations of alpha power and E:I balance, as captured by EEG spectral exponents (*Kosciessa et al., 2021*). Here, task difficulty remained constant throughout the experiment an fluctuations between both modes might not follow momentary demand (*Kosciessa et al., 2021*; *Pettine et al., 2021*) but varying sensory-cognitive resources.

Additionally, attention-related modulations of both alpha power and EEG spectral exponents appeared uncorrelated across individuals – further evidence that they reflect separate neural sources. Future studies that combine a systemic manipulation of E:I (e.g., through GABAergic agonists) with the investigation of attentional load in humans are needed to specify with greater detail how thalamic activity modes drive alpha oscillations and EEG spectral exponents. Specifying potential demand- and resource-dependent trade-offs between different modes of attention-related modulations of cortical activity and sensory processing will offer crucial insights into the neural basis of adaptive behaviour.

## EEG spectral exponents track the 1/f features of sensory stimuli

Over and above the demonstrated anaesthesia- and attention-related effects and their putative relation to E:I balance, we also examined if and how EEG spectral exponents contain information about stimulus characteristics of incoming sensory information. This would reflect a separate neural mechanism that the spectral exponent can come to reflect: A neural tracking of stimulus statistics.

For both modalities, the tracking of stimulus spectral exponents, based on single trials, represented a strong effect, as indicated by standardized effect size estimates (Cohen's d > 0.79). This set of results is especially striking given that conventional sensory-evoked responses or estimates of low-frequency power showed no link with the spectral exponent of sensory stimuli (see *Figure 4—figure supplements 1 and 6*). Furthermore, potential influences of single trial performance on stimulus tracking were ruled out by a control analysis that accounted for single-trial behavioural performance and replicated the results summarized above (see *Figure 4—figure supplement 7*).

The neural tracking of stimulus spectra displayed distinct central and occipital topographies for auditory and visual stimuli, respectively (*Figure 4*). Of course, current source density transforms of sensor-level EEG topographies as used here do not represent definitive evidence for specific cortical sources. Yet, the spatial distinctiveness of both tracking patterns strongly suggests separate cortical origins. Furthermore, the topographies of stimulus tracking strongly resemble those of sensory processing in auditory and visual sensory cortical areas, respectively (*Iemi et al., 2019*; *Waschke et al., 2019*).

These results are conceptually in line with findings from extracellular recordings in ferrets demonstrating the tracking of different AM spectra along the auditory pathway (*Garcia-Lazaro et al., 2006*; *Garcia-Lazaro et al., 2011*), and also extend previous work that analysed oscillatory human brain activity during the presentation of 1/f stimuli (*Hermes et al., 2015*; *Teng et al., 2018*). However, by presenting first evidence for a linear relationship between 1/f-like stimulus features and 1/f-like EEG activity in humans, our results argue for a sensory-specific tracking of AM stimulus exponents within sensory cortical areas at the level of single trials.

Of note, auditory stimulus tracking increased significantly when participants focused their attention on auditory stimuli. A weaker if not statistically significant effect was discernible in the visual domain (*Figure 4B*). Hence, the selective allocation of attentional resources yielded improved tracking of 1/f--like sensory features. Due to too short, stimulus-free inter-trial intervals, we were unable to analyse if the degree of attention-induced reduction in EEG spectral exponents directly reflected the magnitude of stimulus tracking. Future research is needed to further investigate the precise link between

individual averages of EEG spectral exponents, their attention-related change, and the tracking of environmental 1 /f distributed inputs.

## Neural processes potentially driving 1/f stimulus tracking in the EEG

What might constitute the mechanism that, at the level of sensory neural ensembles, gives rise to the observed link between sensory stimuli and the spectral shape of the EEG? First, it is important to emphasise that the representation of stimulus spectra in the EEG does not trace back to an alignment of true oscillatory neural activity and oscillatory stimulus features, commonly referred to as 'entrainment' in the narrow sense (*Obleser and Kayser, 2019*) entrainment in the narrow sense would be contingent on the presence of true endogenous oscillatory activity within recorded EEG that progressively synchronizes its phase to the phase of exogenous stimulus oscillations. Here, presented stimuli were stochastic in nature and without clear sinusoidal signals, preventing narrow-sense oscillatory entrainment from taking place. Neural tracking of the statistical properties of random noise time-series might emerge via the temporal alignment of high amplitude stimulus periods with high amplitude neural activity periods, a mechanism similar to the one implied in the generation of steady-state evoked potentials (SSEPs; *Norcia et al., 2015*). SSEPs are commonly studied by presenting rhythmic sensory input to participants which is assumed to evoked trains of evoked responses at the frequency of presentation, resulting in peak in the EEG spectral at that frequency. However, the stimuli and analysis approaches used in the current study suggest a mechanism of neural tracking that goes beyond common steady state responses to a single presentation frequency.

First, phase and amplitude time-courses of stimuli were dissimilar across trials, preventing phase-locked evoked activity, a hallmark of SSEP generation (*Vialatte et al., 2010*). Second, and unlike most steady state experiments, the tracked stimulus information (AM spectral exponents) was not constant across time but only accessible via temporal integration. Finally, we excluded EEG signals during the first 600 ms after stimulus onset from all tracking analyses and additionally revealed a null-effect of stimulus spectral exponents on single trial evoked potentials. Hence, the observed neural tracking of AM spectral exponents does not emerge via a neural adaptation to constant amplitude spectra or trial-wise differences in evoked responses.

Importantly, the temporal alignment of broadband sensory input with human brain activity has been studied in the context of 'neural tracking' using multivariate linear models and might be able to explain the link between stimulus and EEG spectral properties we observe (*Lalor and Foxe, 2010*; *Wöstmann et al., 2016*). Here, a linear relationship between time-courses of stimulus features and neural responses is assumed to capture their temporal alignment, commonly referred to as 'entrainment in the broad sense' (*Obleser and Kayser, 2019*).

As outlined above, we estimated auditory and visual TRFs to test whether forward modelling of EEG data would result in EEG spectra that mimicked properties of stimulus spectra. However, auditory TRFs were unreliable (see *Figure 4—figure supplement 1*). Visual TRFs on the other hand enabled significant stimulus reconstruction and were used within a simplified proof-of-concept model to predict EEG signals that indeed mimicked the spectral properties of stimuli (*Figure 4—figure supplement 4*). The non-predictiveness of auditory TRFs potentially traces back to an insufficient signal-to-noise ratio and limited training data. In general, EEG spectral exponents might also capture the consequences of non-linear interactions between stimulus input and neural response by focusing on their spectral representation across a wide frequency range. Such non-linear links of stimulus and response are by design inaccessible to TRF approaches that rely on the linear relationship of both time series.

Although spectral-based approaches of neural stimulus tracking clearly displayed higher power in context of the analysed dataset, we deem it probable that both approaches eventually capitalize on the same aspect of central neural processing: the temporal alignment of high amplitude/salience stimulus events with high amplitude neural activity. While this does not correspond to entrainment in the narrow sense or SSVEP-like superposition of oscillatory activity or ERPs, 1 /f AM spectra might evoke trains of evoked responses with similar spectral exponents. Indeed, a simple proof-of-concept model based on real stimulus data resulted in EEG spectra whose exponents were positively linked with the exponents of stimuli (*Figure 4—figure supplement 5*). Hence, time- and spectrally based approaches of stimulus tracking might indeed capture similar aspects of postsynaptic neural activity that align with sensory input during early processing. Importantly, however, future studies are needed to further test

the relationship between temporal neural tracking using TRF approaches and spectral tracking as put forward in the current manuscript.

## Neural stimulus tracking explains inter-individual differences in performance

We used partial least squares to investigate the multivariate between-subject relationship of neural stimulus tracking to behavioural performance. While the statistical power of this between-subject analyses clearly is inferior to other analyses of the current manuscript, the resulting positive correlations (non-parametric) between stimulus tracking and performance cannot be explained by potential outliers (see *Figure 5*). Importantly, this effect was confined to each modality; while individuals who displayed high auditory tracking also displayed fast responses in auditory trials, they exhibited slower and less accurate responses on visual trials. Furthermore, participants who showed strong tracking of visual stimuli performed especially fast and accurate on visual but not auditory trials. This specificity of behavioural benefits through stimulus spectral tracking to each modality argues against the idea of attention-dependent sensory filters that entail bi-directional effects (i.e., auditory attention = visual ignoring; *Lakatos et al., 2013*; *Obleser and Kayser, 2019*).

The sample size of N = 24 is modest (although it does not stand out as small when compared with usual practices of the field), and we have employed non-parametric correlations combined with a two-stage permutation approach to not rely on unwarranted assumptions. Nevertheless, data from a larger cohort of individuals would be ideal to test this between-subject relationship, better estimate its effect size, and test the generalizability of observed neuro-behavioural correlations. Furthermore, introducing within-subject manipulations of difficulty in a comparable design might allow for the investigation of within-subject effects of neural stimulus tracking on perceptual performance.

Of note, the topography of the between-subject neuro-behavioural correlations does not display peaks at the central or occipital regions that were found to show significant stimulus tracking. However, a difference between both topographies is plausible for at least two reasons. First, the sensory stimulus tracking topographies represent fixed effects, by definition minimizing between-subject variance. In contrast, the between subject correlation of stimulus tracking and performance seeks to maximise between-subject variance, increasing the probability that a non-identical topography may be found. Second, although stimulus exponents are significantly tracked at sensor locations that point to early sensory cortices, our multisensory task required more abstract, high-level representations of sensory input for accurate performance. Indeed, the fronto-central topography of the between subject correlation is suggestive of sources in frontal cortex, which have been shown to track multi-sensory information (*Ghazanfar and Schroeder, 2006*; *Senkowski et al., 2007*). Furthermore, prefrontal cortex activity that gives rise to highly similar frontal topographies (*Figure 5*) has been found to represent information about the frequency content of auditory, visual, and somatosensory stimuli (*Spitzer and Blankenburg, 2012*). Thus, the positive link between neural stimulus tracking and performance at fronto-central electrodes points to the behavioural relevance of higher-level stimulus features represented in a supramodal fashion.

## Limitations and next steps

First, attention-dependent changes in EEG spectral exponents might trace back to altered sensory-evoked responses. We argue that such a link is unlikely since differences in evoked responses were limited to the visual domain, occipital electrodes, and an early time-window (80–150 ms post noise onset, see *Figure 3—figure supplement 1*) that was well detached from the time-window used to extract single trial EEG spectra (starting at 600 ms post noise onset).

Additionally, TRF-based models of EEG rather speak to a steepening of spectra via increases of sensory processing that accompany attentional focus, contrary to the observed decreases (see *Figure 4—figure supplement 5*). However, this does not rule out entirely a remaining conflation of selective-attention effects and sensory-processing signatures in the EEG spectral component, as no trials without sensory input were included. Although sensory input was comparable across different attention conditions (auditory and visual stimuli simultaneously), future studies are needed to further specify the link between modality-specific attention and EEG spectral exponents in the absence of sensory input.

Second, one reason for the difference in attentional improvement of stimulus tracking between modalities might lie in the difficulty of the task. Although auditory and visual difficulty were closely matched, we found significantly lower performance for auditory compared to visual trials (see *Figure 2*). Although we deem it unlikely that the observed difference of 3 % in accuracy (70% vs. 73%) might be indicative of a meaningful difference in performance, we cannot rule out the possibility that participants needed more cognitive resources to perform the auditory task and neurally track stimulus spectra. Due to these increased demands, the effects of selective attention might have been able to amplify stimulus tracking more strongly as compared to the potentially less demanding visual condition. Future studies should investigate the role of parametric task demands for stimulus tracking, and attentional improvements thereof, by additionally recording fluctuations in pupil size during constant light conditions as a proxy measure of demand-related fluctuations in arousal (*Yerkes and Dodson, 1908*; *Zekveld et al., 2010*).

## Conclusion

The present data show that the EEG spectral exponent represents a non-invasive approximation of intra-individual variations in states of E:I balance – may these E:I states be driven globally (here, by central-acting anaesthetics) or more focally by re-allocation of selective attention. In addition to these links between E:I and EEG spectral exponents, we highlight the sensitivity of the EEG spectral exponent to aperiodic, 1/f-like stimulus features simultaneously in two sensory modalities and in relation to behavioural outcomes. These findings pose a tightening link from invasive non-human animal physiology to human cognitive neuroscience. They set the stage for a new line of experiments, using non-invasive approximations of aperiodic neural activity to study intra-individual variations in brain dynamics and their role in sensory processing and behaviour.

## Materials and methods
### Pre-processing and analysis of EEG data under different anaesthetics

To test the effect of different central anaesthetics on 1 /f EEG activity, we analysed a previously published openly available dataset (*Sarasso et al., 2015*). Sarasso and colleagues recorded the EEG of healthy individuals during quiet wakefulness and after the administration of different commonly used central anaesthetics including propofol and ketamine. Details regarding the recording protocol can be found in the original study and a recently published re-analysis (*Colombo et al., 2019*; *Sarasso et al., 2015*). We analysed EEG (60 channels) recordings from 10 participants who either received propofol or ketamine infusion (5/5). EEG data were re-referenced to the average of all electrodes, down-sampled to 1000 Hz, and filtered using an acausal finite impulse response bandpass filter (0.3– 100 Hz, order 127). Next, to increase the number of samples per condition, recordings were split up into 5 s epochs. Since the duration of recordings varied between participants, this resulted in different numbers of epochs per anaesthetic and participant (propofol: 98 ± 88 epochs; ketamine: 69 ± 29 epochs). The power spectrum of each epoch and electrode between 1 and 100 Hz (0.25 Hz resolution) was estimated using the Welch method (*pwelch* function). The spectral parameterization algorithm (version 1.0.0; *Donoghue et al., 2020*) was used to parameterize neural power spectra. Settings for the algorithm were set as: peak width limits: [1 – 8]; max number of peaks: 8; minimum peak height: 0.05; peak threshold: 2.0; and aperiodic mode: 'fixed'. Power spectra were parameterized across the frequency range 3–55 Hz.

To statistically compare EEG spectral exponents between quiet wakefulness (resting state) and anaesthesia despite the low number of participants (four per anaesthesia condition), we focused on five central electrodes (see inset in *Figure 1*) and employed a permutation-based approach. After comparing average spectral exponents of resting state and anaesthesia recordings using two separate paired t-tests, we permuted condition labels (rest vs. anaesthesia) and repeated the statistical comparison 1,000 times. Hence, the percentage of comparisons that exceed the observed t-value represents an empirically defined p-value. Note that spectra were normalized before visualization (*Figure 1*) and non-normalized power spectra can be found in the supplements (*Figure 1—figure supplement 1*).

## EEG spectral exponents during a multisensory detection task

To investigate the dynamics of 1 /f EEG activity during varying selective attention and the processing of sensory stimuli with distinct 1 /f features, we recorded EEG from 25 healthy undergraduate students (21 ± 3 years old, 10 male) while they performed a challenging multisensory detection task. All participants gave written informed consent, reported normal hearing and had normal or corrected to normal vision. All experimental procedures were approved by the institutional review board of the University of California, San Diego, Human Research Protections Program. Due to below-chance level performance, one participant had to be excluded from all further analyses.

## Task design and experimental procedure

The novel multisensory design used in the current study required participants to focus their attention to one modality of concurrently presented auditory and visual noise stimuli to detect brief sinusoidal amplitude variations of the presented noise (*Figure 2A*). Participants were asked to press the spacebar as fast and accurately as possible whenever they detected such a sinusoidal AM (target) in the currently attended sensory domain. The experiment was divided into 12 blocks of 36 trials (432 trials total). At the beginning of each block participants were instructed to detect targets embedded in either auditory or visual noise stimuli. The to be attended modality alternated from block to block and was randomized across participants for the first block. Prior to each trial, the central white fixation cross changed its colour to green and back to white to indicate the start of the next trial. After 500 ms the presentation of noise in both modalities started simultaneously. Trials lasted between 4 and 4.5 s, ended with the central fixation cross reappearing on the screen, and were separated by silent inter-trial intervals (uniformly sampled between 2 and 3.25 s). After each experimental block, participants received feedback in the form of a percentage correct score and were asked to take a break of at least 1 min before continuing. Participants were seated in a quiet room in front of a computer screen. The experiment, including EEG preparation, lasted approximately 2.5 hr.

To ensure that the task was challenging but executable for all participants to a comparable degree, we combined training with an adaptive tracking procedure. In detail, participants performed four practice trials of each modality during which target stimuli were clearly detectable. Subsequently, participants performed 12 blocks of 36 trials each during which difficulty was adjusted by changing the modulation depth of presented targets to keep performance constantly around 70 % correct.

## Stimulus generation

Auditory and visual stimuli of different AM spectra were built in three steps: First, 30 s segments of white noise (sampling frequency 44.1 kHz) were generated and high-pass filtered at 200 Hz. Second, four random time-series of the same duration but differing $1/f^\chi$ exponent ($\chi$ = 0, 1, 2, or 3) were generated using an inverse Fourier transform and lowpass filtered at 100 Hz. Finally, separate multiplication of the white noise carrier with the modulators of different spectral exponents resulted in four signals that only varied in their AM but not in their long-term frequency spectra (see *Figure 2B*). The same noise was used for auditory and visual stimuli after root mean square normalization (auditory) or down-sampling to 85 Hz and scaling between 0.5 and 1 (visual). Noise stimuli presented during the experiment were cut out from the 30 s long time-series. Importantly, the AM spectra of cut out noise snippets do not necessarily overlap with the AM spectra of the longer time-series they were cut from. This difference between global and local spectra resulted in a wide distribution of AM spectra that were presented throughout the experiment. AM exponents were uncorrelated between modalities across trials. Auditory noise was presented as amplitude modulated white noise over headphones whereas visual noise was shown as luminance variations of a visually presented disk. Targets consisted of short sinusoidal AMs (6–7.5 Hz, 400 ms) and modulation depth was varied throughout the experiment to keep performance around 70 % correct. All stimuli were generated using custom Matlab code. Auditory stimuli were presented over headphones (Sennheiser using a low-latency audio sound-card (Native Instruments)). Visual stimuli were presented on a computer screen (85 Hz refresh rate). Both auditory and visual stimuli were presented using MATLAB and Psychophysics toolbox (*Brainard, 1997*). To later analyse the relationship between EEG activity, behaviour and the AM spectra of presented stimuli, single trial stimulus spectra were extracted (1–30 Hz, 0.1 Hz resolution, *pwelch* in MATLAB) and parameterized to fit 1 /f exponents (*Donoghue et al., 2020*). Settings for the algorithm were set as: peak width limits: [0.5–12]; max number of peaks: infinite; minimum peak height: 0; peak

threshold: 2.0; and aperiodic mode: 'fixed'. Power spectra were parameterized across the frequency range 1–25 Hz.

## EEG recording and pre-processing

64-channel EEG was recorded at a sampling rate of 1000 Hz using the brainamp and the actichamp extension box (active electrodes; Brainproducts). Artifacts representing heartbeat, movement, eye blinks or saccades and channel noise were removed using independent component analysis based on functions from the fieldtrip and EEGlab toolboxes (*Delorme and Makeig, 2004*; *Oostenveld et al., 2011*). Components were rejected based on power spectra, time-series, topography and dipole fit. Continuous EEG signals were referenced to the average of all channels and filtered between 0.05 and 100 Hz (acausal FIR filter, order 207). Data were segmented into trials between –1 and 5 s relative to trial start (noise onset) and baseline corrected to the average of 1 s prior to trial start. Trials containing artifacts were removed based on visual inspection (5 ± 7 trials rejected). EEG time-series were transformed to scalp current densities using default settings of the fieldtrip toolbox (*ft_scalpcurrentdensity*). Single trial power spectra between 1 and 100 Hz (0.5 Hz resolution) were calculated using the welch method (*Welch, 1967*). To minimize the impact of early sensory-evoked potentials, these spectra were based on the EEG signal between 600 ms after noise stimulus onset and the appearance of a target sound. Trials during which the target appeared within 500 ms after trial start were excluded (1.5% ± 0.3% of trials). Furthermore, mirrored versions of single trial data were appended to the beginning and end of each trial before calculating spectra, effectively tripling the number of samples while not introducing new information (i.e., 'mirror padding'). To approximate EEG activity in a state of awake rest despite the absence of a dedicated resting state recording, we calculated spectra based on the 500 ms before the fixation cross changed its colour, signalling the start of the next trial (same settings as above). These single trial 'resting-state' spectra were averaged to reveal one resting state spectrum per participant and electrode. To estimate 1 /f spectral exponents of EEG activity as well as oscillatory activity, single trial and average resting state spectra were fed into the spectral parameterization algorithm (*Donoghue et al., 2020*) and exponents were fit between 3 and 55 Hz using Python version 3.7. Trials where fits explained less than 20 % of variance in EEG spectra were excluded from all further analyses (0.4 % ± 0.6 % of trials). Note that single trial parameterization provided good fits in all subjects and at all electrodes. *Figure 4—figure supplement 2* provides an overview of fit statistics and examples of two representative single trial fits. To control for the influence of alpha-oscillations (8–12 Hz), we extracted single-trial, single-electrode power estimates from spectral parameterization results if an oscillation was detected within the alpha frequency range (see clear alpha-range peak in Fig. S4). For trials where this was not the case, spectral power was averaged between 8 and 12 Hz as a substitute.

## Statistical analysis

### Attention tracking

To test whether the allocation of attentional resources to one sensory domain is accompanied by a selective flattening of the EEG power spectrum over related sensory areas, we followed a two-step approach. First, we used multiple linear regression to control single trial, single electrode EEG spectral exponents for a number of covariates. Specifically, and for every participant, we controlled EEG spectral exponents for the influence of auditory stimulus exponents, visual stimulus exponents, alpha power, and trial number. Next, we averaged the residuals per electrode and attention condition (auditory vs. visual attention) across trials, resulting in 2 × 64 EEG spectral exponent estimates per participant. The topographical pattern of the average difference between auditory and visual attention EEG spectral exponents is visualized in *Figure 3A*. Following our hypothesis of sensory specific, attention-related flattening of EEG spectra, we averaged EEG spectral exponent residuals across a set of fronto-central (FC1, FC2, Fz, C1, C2, Cz) and parieto-occipital (P03, PO4, POz, O1, O2, Oz) electrodes to contrast activity from auditory and visual sensory areas, respectively. We modelled EEG spectral exponent as a function of ROI (central vs. occipital), attentional focus (auditory vs. visual) and their interaction, including a random intercepts and random slopes for all predictors within a linear mixed model (*fitlme* in MATLAB). Subsequently, analysis of variance (ANOVA) was used to statistically evaluate the main effect of attentional focus as well as the interaction of attentional focus and ROI. Note that attentional focus only varied between blocks of 36 trials, in contrast to stimulus exponents

which varied on a trial-wise basis. Hence, a single-trial modelling approach as described below for stimulus tracking is not warranted here as it would artificially increase statistical power. Furthermore, such an approach would have complicated the statistical evaluation of the hypothesized interaction of ROI with attentional focus.

As an additional control analysis, we tested the subject- and electrode-wise differences of EEG spectral exponents between auditory and visual attention against zero using a cluster-based permutation approach (*Maris and Oostenveld, 2007*). Finally, we used paired t-tests to compare attention effects between ROIs and resolve the interaction effect.

### Stimulus tracking

To test the link between EEG spectral exponents and AM spectral exponents in the auditory and visual domain on the level of single trials, we used single-electrode linear mixed effect models. Model fitting was performed iteratively and hypothesis-driven, starting with an intercept only model and gradually increasing model complexity to find the best fitting model (*Tune et al., 2018*; *Waschke et al., 2019*). After every newly added fixed effect, model fits were compared using maximum likelihood estimation. Once the final set of fixed effects was determined, a comparable procedure was used for random effects. All continuous variables were z-scored across participants (but within electrodes) before entering models, rendering the regression coefficients β direct measures of effect size. The winning model included fixed effects for auditory stimulus exponents, visual stimulus exponents, attentional focus, trial number and resting state EEG spectral exponent (between subject factor) as well as a random intercept for all predictors and random slopes for attentional focus (see *Supplementary files 1 and 2*). To additionally test the role of selective attention for the tracking of the presented sensory stimuli, we modelled separate interactions between auditory as well as visual stimulus exponents with attention, respectively. As models were fit for single electrodes (64 models), we corrected the resulting p-values for multiple comparisons by adjusting the family-wise error rate using the Bonferroni-Holm correction (*Groppe et al., 2011*; *Holm, 1979*).

To arrive at single subject estimates of stimulus tracking, we chose a stepwise regression approach since including random slopes for auditory or visual stimulus exponents within the final mixed models did not improve model fit but led to convergence issues due to model complexity. Hence, on the level of single participants and electrodes, we regressed EEG spectral exponents on attentional focus, trial number, and stimulus exponents of one modality (e.g., visual). The z-scored residuals of this multiple regression were used in a second step where they were regressed on the z-scored stimulus exponent of the remaining sensory domain (e.g., auditory). The resulting beta coefficients were averaged across the electrodes that showed significant stimulus tracking within the mixed model approach, representing single subject estimates of stimulus tracking (see *Figure 3*). By limiting data to trials from one attention condition, single subject estimates of stimulus tracking for different targets of selective attention were calculated following a similar approach.

### Model comparisons

To compare the reported tracking of stimulus exponents in EEG spectral exponents with other established spectral measures of EEG activity and estimate the specificity of the reported effects, we performed formal model comparisons. To this end, we inverted all models and instead of modelling EEG spectral exponents used auditory or visual stimulus exponents as dependent variables. Predictors were identical to the previously reported models (see *Supplementary files 1 and 2* for details) but additionally included either single trial estimates of alpha power (8–12 Hz), low-frequency power (1–5 Hz), or EEG spectral exponents. Note that alpha power estimates were extracted using the same spectral parameterization approach that was used to estimate spectral exponents. Trials without a detected oscillation in the alpha range were excluded from all models to render likelihood comparisons interpretable (11.2% ± 3.4 % of trials excluded). Since oscillations were only seldomly detected in the low-frequency range, we instead used single trial power averaged across this range as a predictor. For each electrode, 4 likelihood ratio tests were performed, one for each stimulus modality and one for each predictor (low-frequency or alpha power), always testing against the respective EEG spectral exponent-based model.

## Temporal response functions

To explore an alternative model of stimulus tracking that operates in the time-domain, we estimated multivariate temporal response functions (mTRFs) of EEG recordings to continuous auditory and visual sensory input. Here, we used the amplitude envelope onsets of auditory as well as visual stimulus time-courses to estimate TRFs. To this end, we filtered the absolute values of their analytic signals below 20 Hz, calculated the first derivative and applied half-wave rectification, down-sampling to 250 Hz, and trial-segmentation, following common standards in the field (*Crosse et al., 2016*; *Fiedler et al., 2019*). EEG data were down-sampled to 250 Hz and cut into trial segments. EEG and stimulus data from all trials were used to train forward or backward multivariate linear models using ridge regression and a set of different regularization parameters ($10^{-3}$–$10^{3}$) as implemented in the mTRF toolbox (*Crosse et al., 2016*).

The forward model approach resulted in auditory and visual response functions per electrode and subject (see *Figure 4—figure supplement 3*). Next, using backward models and leave-one-subject-out cross-validation, we reconstructed stimulus time-courses based on response functions and recorded EEG. To this end, response functions and model constants were averaged across all but one left-out subject after which these TRFs and EEG data from the hold-out subject were used to reconstruct stimulus time-courses. Reconstructed and observed stimuli were correlated on a single trial basis. For each participant, we separately tested the across-trial distributions of correlations against zero using one sample t-tests (see *Figure 4—figure supplement 4*).

Given non-significant stimulus reconstruction of auditory backward models, we could not definitely conclude that phase locking explains the mapping of EEG and stimulus spectra we report. To nevertheless test this hypothesis at least on a conceptual level, we decided to base the prediction of EEG data on a canonical response function (grand average visual TRF) instead of subject- and electrode-wise TRFs. Hence, we predicted one channel EEG data solely based on one canonical TRF per attention condition and trial-wise stimulus time-courses of both modalities. To extract estimates of simulated EEG spectral exponents, we applied the same approach outlined above for real EEG data. The results of this simplistic model were evaluated using a linear model that included main effects for stimulus exponent (four bins of equal size) and attention (auditory/ visual) and their interaction.

## Neuro-behavioural correlation

To investigate whether inter-individual differences in neural stimulus tracking relate to inter-individual differences in performance, we analysed correlations between single subject estimates of stimulus tracking and different metrics of behavioural performance using a multivariate PLS approach (*Krishnan et al., 2011*; *McIntosh et al., 1996*). In brief, so-called 'behavioural PLS' begins by calculating a between-subject correlation matrix linking brain activity at each electrode with behavioural measures of interest. The size of this rank correlation matrix is determined by the number of electrodes, brain variables and behavioural variables [size = ($N_{electrodes} \times N_{brain\ variables}$)$\times N_{behavioural\ variables}$]. In the present study, we used two brain variables with 64 electrodes each (auditory and visual tracking betas) and four behavioural variables (auditory and visual accuracy and response speed). Next, this correlation matrix is decomposed using singular value decomposition (SVD), which results in $N_{brainvar} \times N_{behavvar}$ latent variables (eight in our case).

This approach produces two crucial outputs: (1) A singular value for every latent variable, representing the proportion of cross-block covariance accounted for by that latent variable, and; (2) a pattern of weights (n = number of electrodes) or saliencies representing the correlation strength between stimulus tracking and the used behavioural measures. The multiplication (dot product) of these weights with electrode-wise tracking estimates yields so-called 'brain scores', which here reflect the between-subject relationship of stimulus tracking and performance where positive brain scores indicate that individuals with stronger tracking display better performance. Statistical significance of brain scores and latent variables was tested through permutations of behavioural measures across individuals (5,000 permutations). Additionally, the robustness of weights (saliencies) was estimated using a bootstrap procedure (5,000 bootstraps, with replacement). The division of each weight by the corresponding bootstrapped standard error yields bootstrap ratios, which estimate the robustness of observed effects on an electrode-wise basis. Bootstrap ratios can be interpreted as a pseudo-Z metric. Crucially however, because multivariate PLS is run in a single mathematical step that includes (and weights the importance of) all elements of the brain-behaviour matrix, multiple comparisons

correction is neither typical nor required (*McIntosh et al., 1996*). Furthermore, bootstraps were used to estimate 95 % CIs for observed neuro-behavioural correlations.

Of note, behavioural performance metrics were calculated separately for auditory and visual attention trials. Since difficulty was adaptively adjusted throughout the experiment, leading to vastly different target modulation depths across participants, we controlled auditory and visual accuracy for the final modulation depth of the respective domain and used the residuals as a measure of performance. To furthermore exclude influences of learning and exhaustion of response speed (reaction time$^{-1}$) we controlled single trial response times for trial number and used averaged residuals as subject-wise indicators of response speed. Note that such an approach is especially warranted since we were specifically interested in between-subject relationships and hence the association of correlation matrices between neural tracking and performance. In accordance with such a reasoning, and to prevent outliers in our small sample to obscure results, all PLS analysis were performed using spearman correlation.

## Control analyses

To test the impact of attentional focus and AM stimulus spectra on sensory evoked activity, which might potentially confound differences in EEG spectral exponents, we compared ERPs after noise onset. First, we compared noise onset ERPs between auditory and visual attention using a series of paired t-tests, separately for electrodes Cz and Oz. We corrected for multiple comparisons by adjusting p-values for the false discovery rate (*Benjamini and Hochberg, 1995*). Next, on the level of subjects, voltage values were correlated with stimulus spectral exponents (separately for auditory and visual stimuli) across trials, per electrode, time-point and frequency (*ft_statfun_correlationT* in fieldtrip). On the second level, the resulting t-value time-series were tested against zero using a cluster-based permutation approach (*Maris and Oostenveld, 2007*), separately for auditory and visual stimuli. Finally, to rule out task difficulty as a potential confound of stimulus tracking, we re-ran stimulus tracking mixed models including a main effect of single trial performance (correct vs. incorrect) as well as interactions between single trial performance and auditory and visual stimulus spectral exponents, respectively, to control stimulus tracking for performance and difficulty.

## Acknowledgements

LW and DDG are supported by an Emmy Noether Programme grant from the German Research Foundation (to DDG), and by the Max Planck UCL Centre for Computational Psychiatry and Ageing Research. LW was supported by a G.-A. Lienert fellowship. BV is supported by the Whitehall Foundation Grant 2017-12-73, the National Science Foundation Grant BCS-1736028 and the National Institute of General Medical Sciences Grant R01GM134363-01. JO is supported by the European Research Council (ERC-CoG-2014–646696).

## Additional information

### Competing interests

Jonas Obleser: Reviewing Editor for eLife. The other authors declare that no competing interests exist.

### Funding

| Funder | Grant reference number | Author |
|---|---|---|
| Deutsche Forschungsgemeinschaft | Emmy Noether Programme | Leonhard Waschke Douglas D Garrett |
| H2020 European Research Council | ERC-CoG-2014-646696 | Jonas Obleser |
| Max Planck UCL Centre for Computational Psychiatry and Ageing Research | | Leonhard Waschke Douglas D Garrett |

| Funder | Grant reference number | Author |
| --- | --- | --- |
| Whitehall Foundation | 2017-12-73 | Bradley Voytek |
| National Science Foundation | BCS-1736028 | Bradley Voytek |
| National Institute of General Medical Sciences | R01GM134363-01 | Bradley Voytek |

The funders had no role in study design, data collection and interpretation, or the decision to submit the work for publication.

## Author contributions

Leonhard Waschke, Conceptualization, Data curation, Formal analysis, Investigation, Methodology, Software, Visualization, Writing – original draft, Writing – review and editing; Thomas Donoghue, Formal analysis, Methodology, Software, Writing – review and editing; Lorenz Fiedler, Formal analysis, Software, Writing – review and editing; Sydney Smith, Data curation, Investigation, Project administration; Douglas D Garrett, Formal analysis, Investigation, Visualization, Writing – review and editing; Bradley Voytek, Conceptualization, Investigation, Methodology, Resources, Software, Supervision, Writing – original draft, Writing – review and editing; Jonas Obleser, Conceptualization, Investigation, Methodology, Resources, Supervision, Writing – original draft, Writing – review and editing

## Author ORCIDs

Leonhard Waschke http://orcid.org/0000-0002-1248-9259
Thomas Donoghue http://orcid.org/0000-0001-5911-0472
Lorenz Fiedler http://orcid.org/0000-0002-7892-6917
Douglas D Garrett http://orcid.org/0000-0002-0629-7672
Bradley Voytek http://orcid.org/0000-0003-1640-2525
Jonas Obleser http://orcid.org/0000-0002-7619-0459

## Ethics

Human subjects: All participants gave written informed consent, reported normal hearing and had normal or corrected to normal vision. All experimental procedures were approved by the institutional review board of the University of California, San Diego, Human Research Protections Program (UCSD IRB Protocol #150834. ).

## Decision letter and Author response

Decision letter https://doi.org/10.7554/eLife.70068.sa1
Author response https://doi.org/10.7554/eLife.70068.sa2

# Additional files

## Supplementary files

• Transparent reporting form
• Supplementary file 1. Stimulus tracking model electrode Cz.
• Supplementary file 2. Stimulus tracking model electrode Oz.

## Data availability

Data and code has been deposited on OSF and is available via https://osf.io/wyzrg/.

The following dataset was generated:

| Author(s) | Year | Dataset title | Dataset URL | Database and Identifier |
| --- | --- | --- | --- | --- |
| Waschke L, Donoghue T, Fiedler L, Smith S, Garrett DD, Voytek B, Obleser J | 2021 | MAVIS EEG data | https://osf.io/wyzrg/ | Open Science Framework, 10.17605/OSF.IO/WYZRG |

The following previously published datasets were used:

| Author(s) | Year | Dataset title | Dataset URL | Database and Identifier |
|---|---|---|---|---|
| Massimini M, Laureys S | 2017 | Rest EEG recordings in healthy subjects during wakefulness, sleep and anesthesia with ketamine, propofol, and xenon | https://zenodo.org/record/806176#.YJ58gS8euL4 | Zenodo, 10.5281/zenodo.806176 |

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
