## [Decision Letter]

**Acceptance summary:**

This manuscript reports on two separate investigations. In the first, the authors provide novel evidence from two anaesthesia challenges that the slope of the 1/f structure of the power spectrum of the EEG fluctuates in a manner that tracks the presumed excitation: inhibition (E:I) balance of the tissue generating the EEG signal. Next they show that fluctuations in this slope also covary in systematic and modality- and stimulus-specific ways with behavioral performance on a multimodal attention task. These observations have potential foundational implications for how this previously unappreciated component of the EEG can be interpreted in terms of brain physiology and function.

**Decision letter after peer review:**

Thank you for submitting your article "Modality-specific tracking of attention and sensory statistics in the human electrophysiological spectral exponent" for consideration by *eLife*. Your article has been reviewed by 3 peer reviewers, and the evaluation has been overseen by Maria Chait as Reviewing Editor and Barbara Shinn-Cunningham as the Senior Editor. The following individuals involved in the review of your submission have agreed to reveal their identity: Bradley R Postle (Reviewer #1); Jonathan Z Simon (Reviewer #2).

The Reviewing Editor and reviewers have discussed the reviews with one another. We agree that the balance between excitation and inhibition in the cortex is an important and timely topic. While the approach you use to uncover the role of E/I in attention is interesting, unfortunately, for the reasons outlined below the data as they stand do not support the study's conclusions. We feel that this is potentially addressable with a revision and the RE has drafted this to help you prepare a revised submission.

Essential revisions:

(1) The results of Experiment 1, whilst compelling, require a delicate interpretation. In particular, it is difficult to make a clear distinction between different anaesthetics in terms of their effect on brain activity (see references provided by Rev3). Given this and the low N, results thus do not fully support the strong conclusions offered by the authors. We encourage the authors to revise based on the specific comments from Reviewers 1 and 2 including:

(a) addressing the pattern of spectral effects (propofol mostly enhancing frequencies below ~20-30 Hz and spreading α; ketamine suppressing α while enhancing lower and higher frequencies),

(b) justifying why modelling this as a 1/f change is appropriate, and

(c) quantifying the differences between the different awake spectra (there appears to be a large difference in the awake spectra between the anaesthetics conditions).

Please also acknowledge the limitations associated with small N and existing literature highlighted by Reviewer #3.

(2) The authors interpret the findings of Experiment 2, where changing the value of the spectral exponent in the stimulus resulted in a similar change in the value of the spectral exponent of the response, but only for the selectively attended modality, as originating from an attention-driven change in E/I balance. However, an alternative interpretation of the findings is that these effects reflect attention-driven changes to temporal tracking of the stimulus waveform. These concerns are potentially addressable in a revision but it would require an entirely new data analysis involving a thorough investigation of potential temporal tracking of the stimulus waveform and an unambiguous result. There will need to be a visual temporal analysis (a la VESPA) and auditory temporal analysis (a la AESPA) for both the attended and unattended conditions. The part of the response explained would need to be subtracted out first, and then the "spectral-exponent-tracking" analysis would need to be performed on the residual. There may be additional subtleties that arise in that process. Given the successes of AESPA/VESPA/TRFs in the literature, this should be considered a simpler explanation of the observed response patterns than dependence on E:I balance. It's the residual (true response minus response explained by this mechanism) that would still need an explanation, and that might be argued to be explainable by E:I balance.

*Reviewer #1 (Recommendations for the authors):*

Figure 1, it's really hard to see how the slopes change in the way that the authors state. For propofol, visual inspection suggests that the biggest change is a broadening of the α oscillation, such that its inflection starts at a lower frequency and then because the peak is also 'less pointy,' the purple line simply has to fall at a higher rate to catch up with the gray line by ~30 Hz. For ketamine, at the lowest frequencies (lower than α bump) the slope of the green line simply is steeper than the gray, and then again the biggest difference seems to be that the α bump is abolished with ketamine, and so the gray line is then steeper than the green line for the same reason that purple appears to be steeper than gray in propofol plot. Additionally, there's a lot of jitter with ketamine in the 20-60 Hz range. I realize that visual inspection isn't a rigorous way to analyze these data, but on the other hand it's generally preferable for a figure to clearly illustrate the point that the authors are trying to convey. Perhaps the authors should consider accompanying the 'raw' spectra shown here with the same data decomposed into oscillatory vs. aperiodic components, the way that it is done in the Donoghue et al., (2020) paper?

The Discussion section is largely a repetition of what was written in the Intro and/or a restatement of the results with little additional interpretation and contextualization. For example, although it's important to show that α and aperiodic components of the EEG are statistically dissociable, this is only a step toward understanding more fundamental questions such as (a) what are the functions that periodic vs. aperiodic components support? and (b) what underlying factors that give rise to them?

Here are some more specific comments about the Discussion.

"Jointly, these results underscore the importance of 1/f brain activity for perception and behaviour." Don't the authors really mean: "underscore the utility of parameters of 1/f brain activity for studying the neural bases of perception and behavior"? At the end of the day, the major take-home of this paper is that the slope of the 1/f spectrum is a valid index of E:I balance, but it's E:I balance, per se, that is 'important for perception and behavior,' not the slope itself.

"… these results cannot be explained by attention-dependent differences in neural α power (8-12 Hz, Figure 3), commonly interpreted as a marker of top-down guided sensory inhibition." Idling is an important alternative to inhibition that should be acknowledged.

"First, it is important to emphasise that the representation of stimulus spectra in the EEG likely does not trace back to an alignment of oscillatory neural activity and oscillatory stimulus features, commonly referred to as "entrainment" in the strict sense; the presented stimuli were stochastic in nature and without clear sinusoidal signals. However, neurally tracking the statistical properties of random noise time-series might emerge via a mechanism similar to the one implied in the generation of steady-state evoked potentials (SSEPs)." Both of these seem like important points that merit more elaboration. That is, the word "entrainment" tends to be used carelessly and so more detailed and explicit argumentation about why this is NOT an instance of entrainment would be valuable. With regard to SSEPs, specifying some details about this 'implied mechanism' would be helpful. More generally, although entrainment and evoked responses are precisely specified processes that can be shown to be true or not, the same is not true for "tracking," which is just a loose concept that can't be tested and falsified. Can the authors either specify what they mean by "tracking" or else replace it with a more rigorously defined process?

*Reviewer #2 (Recommendations for the authors):*

P. 4, last paragraph: It is somewhat disconcerting to learn in the Results section that the first study uses a publicly available dataset and the second is wholly separate and from data acquired by the authors. This would be be less startling if it were mentioned in the introduction.

Lines 159-160: As written, this sentence seems to implies that the new results of this paper aren't actually new but merely a confirmation of an old result. It would easier on the reader to more clearly distinguish the previous results (with very strong connections to E:I balance?) from the new findings (where the connection to E:I balance is less direct).

Figure 1B: Would the authors consider using the same vertical scale in both graphs? The overall numbers between the two sets are close enough in value that having two different scales can be distracting.

Figure 1B: The inset graphs are missing axis limits (or scale), and there is no definition of their error bars.

L. 196 and elsewhere: incorrect formatting of numbers in scientific notation, e.g. 7^e-6^ instead of 7 x 10^-6^.

L. 189 and following: The description of the stimuli, especially the auditory stimuli is confusing. The phase "to detect regular (i.e., sinusoidal) amplitude variations in streams of amplitude modulated white noise", in the auditory literature would be understood as analogous to "to detect tone pips in noise", but that is not what is meant here. Figure 2 indicates rather that the stimulus temporarily changes from non-sinusoidal amplitude modulated white noise to sinusoidal amplitude modulated white noise.

Figure 2C: Please explain what the circles and lines represent (I presume individual subjects with lines representing identities, but I need toask after seeing Figure 3B).

Figure 3B: Please explain what the circles and lines represent. Do the lines connect the different tasks of the same individuals? The systematic progression of the slopes of the lines seems to indicate that they do not.

Lines 234-235: Getting R^2^ > 0.84 is a real achievement-it speaks very highly of the importance of the spectral exponent.

L. 383: the phrase "and hence" is confusing here. Maybe "even though they"?

L. 419 and Supplemental Figures: There are two supplemental figures labeled as S4 and none as S5. This reference appears to be to the 5th supplemental figure.

Lines 445-475: This section appears to be where the possibility of temporal tracking is meant to be addressed, but it does not accomplish this (instead only justifying that steady-state analysis does not apply here, which is true). Note also to be careful with the word "stationary". A "stationary process" is one with a fixed spectrum and random phases, which seems to be a good description of the stimulus envelopes/contrasts used here.

Lines 576-577. What does "normalized" mean here? Standard usage is a multiplicative rescaling, not mean-centering. [On the other hand, if the mean-centering was performed on the logarithm (or in dB), then that is equivalent to a multiplicative rescaling of the original waveform.]

L. 616 and following: Regarding the visual modulation, why is the acoustic noise, which had been high-passed at 200 Hz before its modulation, downsampled to 85 Hz (which throws away all the carrier information), instead of just applying the 1/f^X^ modulation directly (downsampled to 85 Hz)? Why the extra complication? Or am I just confused by the multiple uses of the word "noise"?

L. 739 and following: I very much appreciate the careful analysis methods employed here.

Figure S1 caption: This caption would be much clearer if it stated that the graphs and data were identical to that shown in Figure 1 except without normalization. (In its current form it seems almost like an example of an item in a change-blindness study.)

Figures 1B and S1B. There seems to be a lot of inter-subject variability in the Awake case between the subjects who used Propofol vs Ketamine (which should have nothing to do with the awake case). Is that an artifact of changes in the axis scaling (or normalization)? It shouldn't matter since the important statistics are changes within subject, but it is a little disconcerting.

*Reviewer #3 (Recommendations for the authors):*

The authors based their correlation analysis on 24 participants. While the authors do argue that bigger sample size and cross-validation could strengthen the results, the authors could do more with the data they have.

For example, they can employ a leave-one-out linear regression approach, or use k-folds

With regards to the ERP analysis, the authors appear to be using a cluster-permutation approach to assess any differences between the conditions. Here they do have to keep in mind that such a mass-univariate approach is biased towards longer-sustained responses that have a wide scalp distribution, than the rather more focal discrete ERP components. Please see refer to the following discussion on this topic.

https://projects.iq.harvard.edu/files/kuperberglab/files/fieldskuperberg_psychophysiology_2020.pdf

Finally, why I am intrigued by the idea of the slope of 1/f as being something rather important, I am still not convinced that it could be a residual of other factors in the EEG, such as changes in slow frequency power, or evoked responses. I think it would be interesting to see how much unique variance the change 1/f can contribute relative to the other measures of the EEG.

---

## [Author Response]

Essential revisions:(1) The results of Experiment 1, whilst compelling, require a delicate interpretation. In particular, It is difficult to make a clear distinction between different anaesthetics in terms of their effect on brain activity (see references provided by Rev3). Given this and the low N, results thus do not fully support the strong conclusions offered by the authors. We encourage the authors to revise based on the specific comments from Reviewers 1 and 2 including:(a) addressing the pattern of spectral effects (propofol mostly enhancing frequencies below ~20-30 Hz and spreading α; ketamine suppressing α while enhancing lower and higher frequencies),(b) justifying why modelling this as a 1/f change is appropriate, and(c) quantifying the differences between the different awake spectra (there appears to be a large difference in the awake spectra between the anaesthetics conditions).Please also acknowledge the limitations associated with small N and existing literature highlighted by Reviewer #3.

We agree that the effects of anaesthetics on synaptic brain activity are complex and are not yet fully understood. While there are of course additional details and complexities relating to the action of anaesthetics, in this work, we sought to highlight a particular and salient aspect of neural activity, that is the shift in 1/f-structure that can be putatively related to the balance of excitatory and inhibitory activity. We also note that our analysis exists in the broader context of several other related studies that have examined state-related shifts in 1/f-like electrophysiological activity, for example in connection with sleep (Lendner et al., 2020; Bódizs et al., 2021), and also in other investigations of anesthesia (Gao et al., 2017; Colombo et al., 2019; Medel et al., 2020).

It is important to emphasize that this analysis and argument is not the central thesis of our manuscript, but rather is intended as a framing of the main experimental results. While there is ample evidence for predominant and distinct effects of propofol and ketamine on cortical E/I balance (Concas et al., 1991; Zhang et al., 2009; Ching et al., 2010; Brown et al., 2011; Deane et al., 2020) we realize that the initial version of our manuscript could have offered a more nuanced view on the topic. Hence, we have revised sections of introduction and discussion to reflect the complexity of anaesthesia-related dynamics, referencing work suggested by Reviewer 3. Furthermore, we explicitly note the limitation that comes with the small N of study 1 which might at least partially persist despite our use of non-parametric, permutation-based methods and the strong consistency of effects.

We appreciate the reviewers’ thoughtful comments on the specific spectral effects of anaesthesia and related modelling choices as well as interpretations. We have expanded relevant sections of results and discussion paragraphs to also address spectral differences in addition to the spectral exponent (a). Importantly, and with regard to (b), we now elaborate in greater detail on the spectral parameterization that is performed, also providing an additional visualization (see response to R1). In brief, we of course acknowledge the diverse effects of anaesthesia on brain activity which manifests in oscillatory as well as aperiodic aspects of the brain signal. As noted above, however, the goal of this analysis was to test for the potency of EEG spectral exponents as a potential marker of E:I balance by contrasting the spectral exponent of propofol and ketamine anaesthesia recordings against awake rest. While there might also be differences in oscillatory power between anaesthesia conditions, it is unclear how they link to E:I balance. More importantly, the analysed spectral exponents represent the 1/f-like aperiodic part of the spectrum after controlling for oscillatory peaks. Hence, we employ a theoretically guided and methodically supported focus on the EEG spectral exponent instead of an extensive characterization of the effects anaesthesia exerts on EEG spectra.

Nevertheless, the updated version of the manuscript offers more details on the observed spectral changes while explicitly noting the reason and suitability of our focus on spectral exponents, grounded in more recent work in this area.

Finally, we compared resting-state spectra between anaesthesia conditions, finding no significant difference between both (p_perm_ = .75). The absence of such a group difference also becomes apparent in Author response image 1 which directly compares the awake state spectral exponents from both anaesthesia groups. In the updated version of the manuscript, we report this comparison of awake spectral exponents and have updated figure 1 to have constant y-axis limits (see response to R2), making it easier to visually compare spectral exponents of awake recordings.

**Author response image 1. sa2fig1:** EEG PSD exponents do not significantly differ during awake rest. EEG PSD exponents are shown for two groups of subjects, one that later received propofol and another that received ketamine anaesthesia. Single dots represent spectral exponents from 5 second long snippets, horizontal lines depict individual averages. Despite higher inter individual differences in the propofol group, no significant difference between these awake EEG PSD exponents of both groups was found (p = .75).

(2) The authors interpret the findings of Experiment 2, where changing the value of the spectral exponent in the stimulus resulted in a similar change in the value of the spectral exponent of the response, but only for the selectively attended modality, as originating from an attention-driven change in E/I balance. However, an alternative interpretation of the findings is that these effects reflect attention-driven changes to temporal tracking of the stimulus waveform. These concerns are potentially addressable in a revision but it would require an entirely new data analysis involving a thorough investigation of potential temporal tracking of the stimulus waveform and an unambiguous result. There will need to be a visual temporal analysis (a la VESPA) and auditory temporal analysis (a la AESPA) for both the attended and unattended conditions. The part of the response explained would need to be subtracted out first, and then the "spectral-exponent-tracking" analysis would need to be performed on the residual. There may be additional subtleties that arise in that process. Given the successes of AESPA/VESPA/TRFs in the literature, this should be considered a simpler explanation of the observed response patterns than dependence on E:I balance. It's the residual (true response minus response explained by this mechanism) that would still need an explanation, and that might be argued to be explainable by E:I balance.

We thank the reviewers for these suggestions. However, we believe that the perceived issues trace back to a misunderstanding of our original analyses for which we apologize and that we try to resolve step by step below. In brief, we report separate main effects of attention and stimulus spectral exponents on EEG PSD exponents, which we interpret as capturing different neural processes. Importantly, we have also re-analysed all data from experiment 2 using the suggested temporal response function approach and report detailed results after clarifying misunderstandings.

“changing the value of the spectral exponent in the stimulus resulted in a similar change in the value of the spectral exponent of the response, but only for the selectively attended modality”:

This is a misunderstanding. The linear relationship between stimulus spectral exponents and EEG spectral exponents (i.e., stimulus tracking) was present for both modalities, and only partially depended on attentional focus (Figure 4A). While we only found significant auditory stimulus tracking for auditory attention trials (see figure 4B), a similar attention×tracking interaction was not present for the visual domain where stimulus tracking was not significantly affected by attentional focus.

“The authors interpret the findings […] as originating from an attention-driven change in E/I balance.”:

This is not our intended interpretation of the stimulus tracking finding and we apologize if the initial version of our manuscript was not clear enough in differentiating interpretations of the main effects of attention (Figure 3) and stimulus tracking (Figure 4). In the original version of our manuscript, we interpreted the main effect of stimulus tracking as potentially originating from the alignment of high amplitude stimulus periods with high amplitude periods of local field activity (originally Discussion section “1/f stimulus tracking as a sign of non-oscillatory steady state-like activity”, now titled “Neural processes potentially driving 1/f stimulus tracking in the EEG”). Importantly, this assumed temporal alignment of stimulus features and neural activity in early sensory cortices can equally account for a positive link between stimulus and EEG spectra (spectral exponent tracking) and temporally resolved stimulus-EEG correlations (temporal response functions, TRF) as used in the speech tracking approaches suggested by reviewer 2. Hence, both techniques offer complimentary views on the same assumed neural process: changes in brain activity that trace back to the alignment of neural firing and features of sensory input. While we had discussed this idea in the context of and in contrast to SSEPs, we realize that a link with common approaches in the area of speech tracking might have been even more fitting. In the updated version of the manuscript, we explicitly link our findings to common TRF approaches.

“However, an alternative interpretation of the findings is that these effects reflect attention-driven changes to temporal tracking of the stimulus waveform.”:

We assume that the reviewers are referring to stimulus tracking effects (figure 4), specifically to the interaction of attention and stimulus input (figure 4B) and want to highlight that we interpret the significant interaction of attentional focus and stimulus tracking exactly along these lines (see lines 541 and following in the Discussion section).

We wish to point out that we interpret the separate main effects of attention and stimulus tracking to result from different neural processes. In contrast to the main effect of stimulus exponents on EEG PSD exponents, the main effect of attention (figure 3) in our view might capture local changes in cortical E/I balance towards excitation that are unrelated to attention dependent changes in stimulus processing. An important role of stimulus processing in this attention-dependent change of 1/f-like EEG activity appears unlikely for at least three different reasons. (1) All relevant statistical models contained attentional focus, stimulus exponents, and the interaction of stimulus exponent with attentional focus as predictors, controlling for shared influences on EEG spectral exponents. (2) All EEG spectra and hence all following analyses were based on a post-stimulus time-window that did not include the evoked response to noise onset and attention-related differences therein. (3) TRF-based EEG modelling confirmed empirically found attention-dependent changes in ERPs which led to an increase instead of a decrease in PSD exponents of modelled EEG data, tracing back to increased low frequency power (see below for details). Hence, it appears improbable that attention-related changes in sensory processing drive the observed reduction of EEG PSD exponents. If anything, such changes in stimulus processing might render our results to underestimate the true size of attention-related spectral flattening.

Temporal response function analyses:

We agree with the reviewers that a promising alternative approach to analyse the tracking of stimulus spectra in EEG data lies within the estimation of multivariate temporal response functions (mTRF), the widely used descendants of AESPA and VESPA. In brief, such an approach estimates the time-resolved linear relationship between one or more stimulus timeseries and concurrent neural activity. After estimating the impulse response function which captures the time-lagged weights that reveal the common sensory response if applied to neural data, this function can be used to either reconstruct stimulus input (backward model) or predict neural recordings (forward model) from the respective other. The correlation of predicted and observed signals offers an intuitive measure of model fit or predictive accuracy.

Here, we used the amplitude envelope onsets of auditory as well as visual stimulus signals to estimate TRFs. To this end, we filtered the absolute values of the analytic signals below 20 Hz before applying half-wave rectification, down-sampling to 250 Hz, and trial-segmentation, following common standards in the field (Crosse et al., 2016; Wöstmann et al., 2017). EEG data were down-sampled to 250 Hz and cut into trial segments. We then used EEG and stimulus data from all trials to train forward or backward multivariate linear models using ridge regression and a set of different regularization parameters (10-3–103) as implemented in the mTRF toolbox (Crosse et al., 2016). This approach resulted in auditory and visual response functions per electrode and subject. As can be discerned from figure 4 supplement 2, visual response functions displayed a common pattern of deflections that is reminiscent of sensory-evoked responses as well as a common occipital topography and were highly consistent across individuals. Auditory response functions, however, only displayed one peak at a lag of approximately 150-200 ms with an uncommon centro-parietal topography and high inconsistency across individuals (compare size of CIs). These results cast doubt on the suitability of the mTRF approach to predict stimuli or EEG signals given the data analysed in the present manuscript.

We continued by utilizing a leave-one-subject-out cross-validation approach to reconstruct stimulus time-courses based on response functions and recorded EEG. Thus, response functions and model constants were averaged across all but one hold-out subject before these response functions and EEG data from the hold-out subject were used to reconstruct stimulus signals. Reconstructed and observed stimuli were correlated on a single trial basis. For each participant, we separately tested the across-trial distributions of correlations against zero using one sample t-tests. While this procedure revealed significant stimulus reconstruction in the visual domain, this was not the case for auditory stimuli (see with figure 4 supplement 3). Given that such backward models are considerably more powerful compared to forward models (predicting EEG) due to the number of used features (64 electrodes across time), these results render interpretable reconstructions of EEG data based on auditory stimuli unlikely. The difference in response function stability and stimulus reconstruction power between auditory and visual modalities might trace back to variations in SNR between central and occipital electrode locations with lower SNR and decreased selectivity for auditory cortical activity at central electrodes as compared to visual cortical activity at occipital sites. Additionally, the employed design always presented stimuli from both modalities concurrently – a major difference to previous studies that showed tracking of noise amplitudes in the EEG (Lalor et al., 2009). In such a multisensory setup, visual inputs might have dominated and superposed auditory inputs at least partially. Considering these results, our finding of significant links between single trial auditory stimulus exponents and EEG PSD exponents speaks to the power of this spectrum-based approach. Single trial EEG spectral exponents might be sensitive to non-linear links between stimulus and postsynaptic activity that temporal domain response function approaches miss.

Given the instability of auditory TRFs and the failure to reconstruct stimulus time-series using backward modelling, we decided to employ a exemplary forward modelling approach to predict EEG signals from TRFs and stimuli. Instead of relying on subject-specific temporal response functions, we utilized two canonical response functions: the grand average visual response functions at electrode Oz from auditory and visual attention conditions (Figure 4- Figure supplement 2). We used these canonical response functions to predict single trial EEG data based on auditory or visual stimuli and calculated spectra of the predicted EEG signals. These spectra were parameterized using the same approach applied to empirical EEG data.

This modelling approach disregards individual differences as well as topographical information. However, it allowed to theoretically test whether the temporal tracking of sensory input in the EEG might result in EEG spectra that mimic the shape of stimulus spectra. As can be discerned from figure 4 supplement 4, predicted spectra displayed the commonly observed 1/f decay in power with increasing frequency, interrupted by a peak around 15 Hz. Although the centre frequency of this peak exceeded the usual range of α oscillations, we deem the spectra of modelled data a satisfactory approximation of human EEG signals.

Importantly, the spectral exponent of predicted EEG signals was positively linked with the spectral exponent of presented amplitude modulation spectra, both for the visual and the auditory domain (beta_vis_ = 0.59, SE = 0.07, t = 8.9, p < 0.0001; beta_aud_ = 0.37, SE = 0.08, t = 4.3, p < 0.0001). These findings illustrate that the temporal alignment of sensory cortical activity with stimulus intensity fluctuations, sometimes referred to as phase-locking or entrainment in the broad sense (Obleser and Kayser, 2019), could in theory account for links between stimulus and EEG spectra we report. However, as outlined above, the data presented in the current manuscript resulted in unreliable individual TRFs and non-significant stimulus reconstruction for the auditory domain. Hence, our data failed to support this hypothesis empirically. This disparity of model and data might be rooted in the insufficient SNR of used EEG recordings or the limited duration of used sensory stimuli. Additionally, common speech tracking approaches rely on at least 20 min of sensory presentation per participant whereas the analysed data only contained roughly 13 min total per participant (13.2 ± 0.27 min). This limited amount of data might be insufficient to train accurate TRFs in the auditory domain.

Nevertheless, the modelling results provide a proof of concept, illustrating that both temporal tracking and exponent tracking analyses eventually might capture the same sensory-related neural process. The used spectral exponent tracking approach, however, appears more powerful in the context of the analysed data as it is able to unearth the effect of stimulus exponents on EEG data were TRF approaches are not. Overall, these results speak to a model in which the temporal alignment of sensory cortical postsynaptic activity with sensory input results in EEG signals that are temporally locked to time courses of stimulus features, resulting in EEG spectra that mimic stimulus spectra. Importantly, PSD-based approaches appeared more powerful in the context of our data, rendering the empirical link between both approaches a hypothesis to be formally tested in future work.

Furthermore, spectral exponent increased when data were modelled based on visual stimuli and the visual attention TRF and vice versa for auditory stimuli (beta_vis_ = .64, Se = .1, t = 5.6, p <.0001; beta_aud_ = .78, SE = .14, t = 5.7, p < .0001). This spectral steepening likely traces back to the increased amplitude of low-frequency parts in the TRF that mimic early ERP effects and is in stark contrast to our empirical finding of topographically specific reduced spectral exponents through modality-specific attention. Thus, attention-related increases of sensory processing might indeed entail an increase in EEG PSD exponents which takes place in parallel with attention-related decreases of exponents that capture a shift in E/I balance. By analysing comparable distribution of stimulus exponents across both attention conditions and statistically controlling for single trial stimulus exponents, the analysis put forward within the present manuscript is able to isolate the attention-related decrease in spectral exponents.

Reviewer #1 (Recommendations for the authors):Figure 1, it's really hard to see how the slopes change in the way that the authors state. For propofol, visual inspection suggests that the biggest change is a broadening of the α oscillation, such that its inflection starts at a lower frequency and then because the peak is also 'less pointy,' the purple line simply has to fall at a higher rate to catch up with the gray line by ~30 Hz. For ketamine, at the lowest frequencies (lower than α bump) the slope of the green line simply is steeper than the gray, and then again the biggest difference seems to be that the α bump is abolished with ketamine, and so the gray line is then steeper than the green line for the same reason that purple appears to be steeper than gray in propofol plot. Additionally, there's a lot of jitter with ketamine in the 20-60 Hz range. I realize that visual inspection isn't a rigorous way to analyze these data, but on the other hand it's generally preferable for a figure to clearly illustrate the point that the authors are trying to convey. Perhaps the authors should consider accompanying the 'raw' spectra shown here with the same data decomposed into oscillatory vs. aperiodic components, the way that it is done in the Donoghue et al., (2020) paper?

It is true that anaesthesia-related spectral changes are not limited to the spectral exponent but rather include several oscillatory alterations as well. Importantly, we account for these changes by parameterizing both oscillatory and 1/f-like signal properties of the EEG. While we understand the reasoning behind the reviewer’s comment on the link between the width of the α peak and the steepness of the power decay immediately above the α range, we want to emphasize that spectral parameterization was performed up to 60 Hz. Hence, a steeper decay after a wider α peak is not sufficient to drive a higher exponent estimate.

To improve the visual presentation of these effects, we followed the reviewer’s suggestion and now plot oscillatory as well as aperiodic fits in addition to average spectra. Note that spectra are normalized by dividing by the mean as suggested by R2. Furthermore, updated results are based on data from 5 participants in each group, including two datasets that we had previously missed (Author response image 2) .

**Author response image 2. sa2fig2:** EEG PSD exponents track anaesthesia-induced E:I changes. (A) Normalized EEG spectra averaged across 5 subjects and 5 central electrodes (inset) displaying a contrast between rest and propofol (left) and ketamine anaesthesia (right). Spectral parameterization yielded aperiodic fits that estimated the spectral exponent (dashed lines) and full fits that included oscillatory spectral peaks (transparent lines). (B) Pairwise scatter plots depicting subject-wise averaged EEG PSD exponents during awake rest, propofol (left) and ketamine (right). Coloured dots represent PSDexponents of 5 s snippets, black horizontal bars single subject means. P-values are based on 1000 random permutations.

On a more general note, regarding the specificity of anaesthesia effects on E/I and related shifts in EEG spectral exponents, we like to point the reviewer to our response to Reviewer 3 below. In brief, we argue that the assumption of differentially altered cortical E/I balance due to propofol or ketamine represents a simplification that is warranted given both previous findings and the goals of the current study. We neither wish to negate the complexity of anaesthesia-related changes in brain activity nor do we want to claim exclusive aperiodic changes due to anaesthesia and E/I. Instead, as nicely summarized by the reviewer above, our goal is to test the assumption of anaesthesia- and E/I-related changes in spectral exponents in non-invasive recordings.

The Discussion section is largely a repetition of what was written in the Intro and/or a restatement of the results with little additional interpretation and contextualization. For example, although it's important to show that α and aperiodic components of the EEG are statistically dissociable, this is only a step toward understanding more fundamental questions such as (a) what are the functions that periodic vs. aperiodic components support? and (b) what underlying factors that give rise to them?

We thank the reviewer for these helpful comments on the Discussion section.

We now discuss the presence of statistically unrelated attentional modulations of both α oscillations and EEG PSD exponents as a potential sign of distinct modes of thalamic firing that shape cortical activity in a demand- and resource-dependent manner. The relevant section is pasted below.

“Despite these differences in the sensitivity of EEG signals, our results provide clear evidence for a modality-specific flattening of EEG spectra through the selective allocation of attentional resources. […] Specifying potential demand- and resource-dependent trade-offs between different modes of attention-related modulations of cortical activity and sensory processing will offer crucial insights into the neural basis of adaptive behaviour.”

We also have re-worked large parts of the discussion aiming to offer our thoughts on potential mechanistic differences and similarities between approaches. Importantly, we now explicitly contrast temporal response function ideas against spectrally-based methods that we put forward in the current manuscript. In brief, we argue that both approaches might very well capture the same alignment of postsynaptic neural activity with stimulus input in different ways. However, while TRF approaches imply linearity and phase locking, the comparison of power spectra and their spectral exponents also incorporates potential non-linear aspects of stimulus response relationships. The relevant section is pasted below.

“Importantly, the temporal alignment of broadband sensory input with human brain activity has been studied in the context of “neural tracking” using multivariate linear models and might be able to explain the link between stimulus and EEG spectral properties we observe (Lalor and Foxe, 2010; Wöstmann et al., 2017). […] Importantly, however, future studies are needed to further test the relationship between temporal neural tracking using TRF approaches and spectral tracking as put forward in the current manuscript.”

Here are some more specific comments about the Discussion."Jointly, these results underscore the importance of 1/f brain activity for perception and behaviour." Don't the authors really mean: "underscore the utility of parameters of 1/f brain activity for studying the neural bases of perception and behavior"? At the end of the day, the major take-home of this paper is that the slope of the 1/f spectrum is a valid index of E:I balance, but it's E:I balance, per se, that is 'important for perception and behavior,' not the slope itself.

The suggested refinement makes sense and we have rephrased accordingly. However, we wish to point out that not all results of the current manuscript can be subsumed under the umbrella of “spectral exponents track E/I”. The alignment of 1/f-like aperiodic stimulus input and aperiodic EEG activity potentially does not represent E/I changes per se but rather an SSEP-like process of temporal stimulus tracking in sensory cortices.

"… these results cannot be explained by attention-dependent differences in neural α power (8-12 Hz, Figure 3), commonly interpreted as a marker of top-down guided sensory inhibition." Idling is an important alternative to inhibition that should be acknowledged.

We now acknowledge the idling hypothesis and cite relevant work.

“Despite these differences in the sensitivity of EEG signals, our results provide clear evidence for a modality-specific flattening of EEG spectra through the selective allocation of attentional resources. This attention allocation likely surfaces as subtle changes in E:I balance (Borgers et al., 2005; Harris and Thiele, 2011). Importantly, these results cannot be explained by observed attention-dependent differences in neural α power (8–12 Hz, Figure 3) which have been suggested to capture cortical inhibition or idling states (Cooper et al., 2003; Pfurtscheller et al., 1996). Also note that the employed spectral parameterization approach enabled to us to separate 1/f like signals from oscillatory activity and hence offered distinct estimates of spectral exponent and α power that would otherwise have been conflated (Donoghue et al., 2020).”

"First, it is important to emphasise that the representation of stimulus spectra in the EEG likely does not trace back to an alignment of oscillatory neural activity and oscillatory stimulus features, commonly referred to as "entrainment" in the strict sense; the presented stimuli were stochastic in nature and without clear sinusoidal signals. However, neurally tracking the statistical properties of random noise time-series might emerge via a mechanism similar to the one implied in the generation of steady-state evoked potentials (SSEPs)." Both of these seem like important points that merit more elaboration. That is, the word "entrainment" tends to be used carelessly and so more detailed and explicit argumentation about why this is NOT an instance of entrainment would be valuable. With regard to SSEPs, specifying some details about this 'implied mechanism' would be helpful. More generally, although entrainment and evoked responses are precisely specified processes that can be shown to be true or not, the same is not true for "tracking," which is just a loose concept that can't be tested and falsified. Can the authors either specify what they mean by "tracking" or else replace it with a more rigorously defined process?

Thank you for these perceptive comments. We have entirely re-structured the paragraph in question to focus more on the comparison of temporal response function approaches that link stimulus and EEG signals in the time domain with PSD exponent-based techniques employed in the current manuscript. Nevertheless, we have added a more detailed explanation of SSEPs and made it clearer why simple steady state responses cannot explain the observed effects.

Additionally, we now provide an explanation of entrainment in the strict sense and outline why it is an unlikely explanation for observed result.

“What might constitute the mechanism that, at the level of sensory neural ensembles, gives rise to the observed link between sensory stimuli and the spectral shape of the EEG? […] Hence, the observed neural tracking of AM spectral exponents does not emerge via a neural adaptation to constant amplitude spectra or trial-wise differences in evoked responses.”

Reviewer #2 (Recommendations for the authors):P. 4, last paragraph: It is somewhat disconcerting to learn in the Results section that the first study uses a publicly available dataset and the second is wholly separate and from data acquired by the authors. This would be be less startling if it were mentioned in the introduction.

We apologize for this unnecessary surprise and now note the use of previously published data in the introduction section.

Lines 159-160: As written, this sentence seems to implies that the new results of this paper aren't actually new but merely a confirmation of an old result. It would easier on the reader to more clearly distinguish the previous results (with very strong connections to E:I balance?) from the new findings (where the connection to E:I balance is less direct).

We have rephrased to make clear that our approach aims at a non-invasive extension and generalization of previous findings.

Figure 1B: Would the authors consider using the same vertical scale in both graphs? The overall numbers between the two sets are close enough in value that having two different scales can be distracting.

We have changed the figure to use the same vertical scale in both panels. Note that spectra are normalized by dividing by the mean as suggested by R2. Furthermore, updated results are based on data from 5 participants in each group, including two datasets that we had previously missed (Author response image 2) .

Figure 1B: The inset graphs are missing axis limits (or scale), and there is no definition of their error bars.

These 45° plots had equal x-axis and y-axis limits and were used with the goal of further illustrating the consistency of anaesthesia effects on EEG PSD exponents. After careful consideration, we decided that they did not help the visibility of effects and have removed them in the updated version of the manuscript.

L. 196 and elsewhere: incorrect formatting of numbers in scientific notation, e.g. 7^e-6^ instead of 7 x 10^-6^.

We fixed that mistake.

L. 189 and following: The description of the stimuli, especially the auditory stimuli is confusing. The phase "to detect regular (i.e., sinusoidal) amplitude variations in streams of amplitude modulated white noise", in the auditory literature would be understood as analogous to "to detect tone pips in noise", but that is not what is meant here. Figure 2 indicates rather that the stimulus temporarily changes from non-sinusoidal amplitude modulated white noise to sinusoidal amplitude modulated white noise.

We rephrased this section and now note that “participants had to detect brief time periods during which the amplitude modulation of the presented white noise switched from aperiodic to sinusoidal.”

Figure 2C: Please explain what the circles and lines represent (I presume individual subjects with lines representing identities, but I need to to ask after seeing Figure 3B).

The reviewer is assuming correctly that dots represent single subject data and lines are taken to depict identities and we have added the relevant information in the figure caption.

Figure 3B: Please explain what the circles and lines represent. Do the lines connect the different tasks of the same individuals? The systematic progression of the slopes of the lines seems to indicate that they do not.

Circles represent single subject residual EEG PSD exponents at central (lilac) or occipital (teal) electrodes, averaged across auditory or visual attention trials. Lines identify individuals.

The systematic progression simply stems from the process of residualization which was performed within subjects, effectively zero-centering individuals. Note that this is not an issue, but desired as it allows us to zoom in on the within subject effects of attentional focus on EEG PSD exponents while controlling for stimulus exponents and other covariates. However, we happily provide an analogous visualization of raw EEG PSD in Author response image 3 . Here, the same attention dependent reduction of PSD exponents as well as its topographical dependence can be seen. However, between subject variance obscures the central within-subject effects.

**Author response image 3. sa2fig3:** Raw EEG spectral exponents change with attention. As in figure 3A, but based on raw EEG spectral exponents instead of model residuals. Individual average exponents are depicted by dots, grand averages by horizontal lines. Coloured lines denote subject identity. Exponents are shown for auditory (left) and visual attention (right) and for a cluster of central (lilac) and occipital electrodes (teal).

Lines 234-235: Getting R^2^ > 0.84 is a real achievement-it speaks very highly of the importance of the spectral exponent.

Thank you for this friendly comment.

L. 383: the phrase "and hence" is confusing here. Maybe "even though they"?

We have rephrased to:

“Central electrodes capture auditory cortical activity but are positioned far away from their dominant source (Huotilainen et al., 1998; Stropahl et al., 2018). Occipital electrodes, however, are sensitive to visual cortex activity and are directly positioned above it (Hagler et al., 2009)”

L. 419 and Supplemental Figures: There are two supplemental figures labeled as S4 and none as S5. This reference appears to be to the 5th supplemental figure.

Sorry for the confusion. We have corrected this mistake and refer to each supplemental figure by linking it with its parent figure in the updated version of the manuscript.

Lines 445-475: This section appears to be where the possibility of temporal tracking is meant to be addressed, but it does not accomplish this (instead only justifying that steady-state analysis does not apply here, which is true). Note also to be careful with the word "stationary". A "stationary process" is one with a fixed spectrum and random phases, which seems to be a good description of the stimulus envelopes/contrasts used here.

We agree that this section in its original form did not achieve what we were aiming for and have edited it to discuss potential neural mechanisms similar to what is commonly referred to as speech tracking. The relevant section is pasted below.

“Importantly, the temporal alignment of broadband sensory input with human brain activity has been studied in the context of “neural tracking” using multivariate linear models and might be able to explain the link between stimulus and EEG spectral properties we observe (Lalor and Foxe, 2010; Wöstmann et al., 2017). Here, a linear relationship between time-courses of stimulus features and neural responses is assumed to capture their temporal alignment, commonly referred to as “entrainment in the broad sense” (Obleser and Kayser, 2019).

As outlined above, we estimated auditory and visual TRFs to test whether forward modelling of EEG data would result in EEG spectra that mimicked properties of stimulus spectra. However, auditory TRFs were unreliable (see Figure 4 supplement 1). Visual TRFs on the other hand enabled significant stimulus reconstruction and were used within a simplified proof-of-concept model to predict EEG signals that indeed mimicked the spectral properties of stimuli (Figure 4 supplement 4). The non-predictiveness of auditory TRFs potentially traces back to an insufficient signal to noise ratio and limited training data. In general, EEG spectral exponents might also capture the consequences of non-linear interactions between stimulus input and neural response by focusing on their spectral representation across a wide frequency range. Such non-linear links of stimulus and response are by design inaccessible to TRF approaches that rely on the linear relationship of both time series.

Although spectral-based approaches of neural stimulus tracking clearly displayed higher power in context of the analysed dataset, we deem it probable that both approaches eventually capitalize on the same aspect of central neural processing: the temporal alignment of high amplitude/salience stimulus events with high amplitude neural activity. While this does not correspond to entrainment in the narrow sense or SSVEP-like superposition of oscillatory activity or ERPs, 1/f AM spectra might evoke trains of evoked responses with similar spectral exponents. Indeed, a simple proof-of-concept model based on real stimulus data resulted in EEG spectra whose exponents were positively linked with the exponents of stimuli (Figure 4 supplement 5). Hence, time- and spectrally-based approaches of stimulus tracking might indeed capture similar aspects of postsynaptic neural activity that align with sensory input during early processing. Importantly, however, future studies are needed to further test the relationship between temporal neural tracking using TRF approaches and spectral tracking as put forward in the current manuscript.”

Lines 576-577. What does "normalized" mean here? Standard usage is a multiplicative rescaling, not mean-centering. [On the other hand, if the mean-centering was performed on the logarithm (or in dB), then that is equivalent to a multiplicative rescaling of the original waveform.]

The reviewer is assuming correctly that mean-centering was performed on the logarithm. For the updated version of the manuscript, we performed multiplicative re-scaling before taking the logarithm of the spectrum.

L. 616 and following: Regarding the visual modulation, why is the acoustic noise, which had been high-passed at 200 Hz before its modulation, downsampled to 85 Hz (which throws away all the carrier information), instead of just applying the 1/f^X^ modulation directly (downsampled to 85 Hz)? Why the extra complication? Or am I just confused by the multiple uses of the word "noise"?

Visual noise was down-sampled to match the maximal refresh rate of the used monitor. You are of course correct in stating that this down-sampling will get rid of carrier information. Importantly, since the visual stimulus consisted of a white disc that fluctuated in luminance according to the used amplitude modulation spectra, carrier information (i.e., frequency spectra) played a very limited role. In other words, while high pass-filtered white noise was used as the carrier signal in the auditory domain, luminance was used as carrier in the visual domain. We realize that our approach might be a bit idiosyncratic but are confident that it resulted in the desired outcome: amplitude modulation spectra with varying 1/fx in the visual domain.

L. 739 and following: I very much appreciate the careful analysis methods employed here.

Thank you.

Figure S1 caption: This caption would be much clearer if it stated that the graphs and data were identical to that shown in Figure 1 except without normalization. (In its current form it seems almost like an example of an item in a change-blindness study.)

We apologize for the confusion and agree that a focus on the difference to figure 1 makes much more sense. We have changed the figure to only show raw power spectra and changed the caption accordingly.

Figures 1B and S1B. There seems to be a lot of inter-subject variability in the Awake case between the subjects who used Propofol vs Ketamine (which should have nothing to do with the awake case). Is that an artifact of changes in the axis scaling (or normalization)? It shouldn't matter since the important statistics are changes within subject, but it is a little disconcerting.

To explore potential differences in awake rest EEG spectra between anaesthetic conditions, we ran analogous analyses to the ones used to test anaesthetic-depended spectral exponent changes. We observed no significant difference between awake rest spectral exponents of both anaesthetic conditions (pperm = .75). The impression of a difference to a large degree was driven by the difference in y-axis scaling which we have changed to equal axes in the updated version of the manuscript.

Reviewer #3 (Recommendations for the authors):The authors based their correlation analysis on 24 participants. While the authors do argue that bigger sample size and cross-validation could strengthen the results, the authors could do more with the data they have.For example, they can employ a leave-one-out linear regression approach, or use k-folds

We thank the reviewer for their comment regarding the potential to strengthen our between subject correlation analyses. However, neither leave-one out nor k-folds cross-validation appear as appropriate analyses choices in this case. One the one hand, leave-one out cross validation has been shown to not provide reliable but noisy and biased results based on samples with less than a couple hundred observations (Varoquaux, 2019). Analogously, foldwise cross validation further reduces the size of training sets by holding out a larger proportion of observations for later testing and hence requires even larger sample sizes to offer reliable results. In the light of these methodical considerations and to account for the small sample size, we chose to apply a two-stage permutation approach which has been developed for the reliable estimation and testing of brain-behaviour relationships in limited sample sizes and is being used widely (McIntosh et al., 1996; Krishnan et al., 2011). The employed partial least squares approach first tests the significance of observed brain-behaviour relationships in latent space by permuting data across participants (here: 1000 permutations). In a second step, a test statistic is estimated using a bootstrap approach (here: 1000 bootstraps). By comparing observed brain-behaviour relationships with mean and standard deviations of bootstrap results, a test-statistic that can be interpreted in line with a z-distribution is generated. In addition to this two-step procedure, we implemented a rank-based version of the entire approach, enabling us to non-parametrically test between-subject relationships of stimulus tracking and performance. Hence, combining a two-stage permutation approach with rank-based correlation, the used methods are well suited for the analyses of the present data.

Nevertheless, we interpret the reported across-subject correlations with great care and refrain from suggesting the prediction of behaviour based on the observed stimulus-related brain dynamics. Furthermore, we acknowledge the importance of generalization and out of sample prediction in the Discussion section and suggest approaches to achieve these goals in future studies.

With regards to the ERP analysis, the authors appear to be using a cluster-permutation approach to assess any differences between the conditions. Here they do have to keep in mind that such a mass-univariate approach is biased towards longer-sustained responses that have a wide scalp distribution, than the rather more focal discrete ERP components. Please see refer to the following discussion on this topic.

https://projects.iq.harvard.edu/files/kuperberglab/files/fieldskuperberg_psychophysiology_2020.pdf

While mass-univariate approaches might not be sensitive to very focal ERP differences, this does not represent an issue in our case for at least three reasons. First, as can be discerned from figure 4 supplement 1, ERPs of different stimulus conditions did not even show slight evidence for focal differences. Second, focal increases of visually-evoked ERPs between auditory and visual attention were detected with the chosen mass-univariate approach (see figure 3 supplement 1), rendering it unlikely that comparably focal stimulus condition differences in the same data might have been missed. Third, and most importantly, the time periods of early evoked responses (0–600 ms post-stimulus onset) were excluded from the estimation of power spectra and hence also the calculation of spectral exponents. Thus, even if the chosen statistical approach missed very focal and faint differences in ERPs, these were unable to affect estimates of EEG spectral exponents and the main results of our study.

Finally, why I am intrigued by the idea of the slope of 1/f as being something rather important, I am still not convinced that it could be a residual of other factors in the EEG, such as changes in slow frequency power, or evoked responses. I think it would be interesting to see how much unique variance the change 1/f can contribute relative to the other measures of the EEG.

We agree with the reviewer on the importance of differentiating the 1/f decay in power and the associated exponent from oscillatory signal parts and related peaks in the spectrum. Importantly, recent work by the second author of the present manuscript (Donoghue et al., 2020), put forward a parameterization approach that achieves exactly this. We have tried to make this clearer by including plots of fitted spectra that differentiate between oscillatory and aperiodic signals in the current manuscript. We wish to refer the reviewer to the original paper for simulation results and an at length discussion of potential links and conflations between oscillatory and aperiodic signal parts. In addition, we want to highlight that 1/f like signals have been argued to play a central role in sensory processing, different forms of cognition, development, and aging (He et al., 2010; Dave et al., 2018; Sheehan et al., 2018; Chini et al., 2021). Importantly, the relevance of 1/f-like parts of brain activity does not hinge on the specific methods employed in the current study but rather persist across imaging techniques from fMRI and M/EEG (He et al., 2010; Dave et al., 2018) towards invasive LFPs (Sheehan et al., 2018) and single unit recordings in non-human animals (Chini et al., 2021).

Regarding the reviewer’s comment on the unique variance accounted for by 1/f exponent changes, we assume that they are referring to variance in behaviour. While not strictly representing overt behaviour as such, selective attention exerted topographically-specific effects on EEG spectral exponents that took place over and above an observed change in α power. Hence, we report unique attention- and behaviourally-linked changes in brain activity that are specific to EEG spectral exponents. Furthermore, analysing inter-individual differences in behavioural performance, we performed a multivariate partial least squares approach to relate the tracking of stimulus exponents in EEG spectral exponents with detection performance. Testing the role of low-frequency power or α power in this analysis requires a significant association between EEG power in the respective frequency bands and stimulus spectral exponents which then can be used to explain inter-individual variance in behaviour. To follow the reviewer’s suggestion (and comments by R1), we calculated the linear relationship of low-frequency and α power with stimulus spectral exponents by reverting mixed models. We compared these models to models of same size that used EEG spectral exponents as a main predictor of interest and was identical otherwise. As can be taken from Figure 4-supplement 6, neither low-frequency nor α power explained significantly more variance in stimulus exponents than EEG spectral exponents for most conditions. Only α power led to increased model fit at one parietal electrode when modelling visual stimulus exponents. Importantly, this electrode did not overlap with the cluster of significant stimulus exponent tracking by EEG exponents and thus likely traces back to a different neural source.